# Current Status and Challenges of Vaccine Development for Seasonal Human Coronaviruses

**DOI:** 10.3390/vaccines13111168

**Published:** 2025-11-16

**Authors:** Bin Zhang, Yaoming Liu, Tao Chen, Jintao Lai, Sen Liu, Xiaoqing Liu, Yiqiang Zhu, Haiyue Rao, Haojie Peng, Xiancai Ma

**Affiliations:** 1Guangzhou National Laboratory, Guangzhou International Bio-Island, Guangzhou 510005, China; zhang_bin@gzlab.ac.cn (B.Z.); liu_yaoming@gzlab.ac.cn (Y.L.); chen_tao02@gzlab.ac.cn (T.C.); lai_jintao@gzlab.ac.cn (J.L.); liu_sen@gzlab.ac.cn (S.L.); liu_xiaoqing@gzlab.ac.cn (X.L.); zhu_yiqiang@gzlab.ac.cn (Y.Z.); rao_haiyue@gzlab.ac.cn (H.R.); peng_haojie@gzlab.ac.cn (H.P.); 2College of Life Science and Technology, Huazhong University of Science and Technology, Wuhan 430074, China; 3Institute of Human Virology, Zhongshan School of Medicine, Sun Yat-sen University, Guangzhou 510080, China; 4School of Biology and Biological Engineering, South China University of Technology, Guangzhou 510006, China; 5State Key Laboratory of Respiratory Disease, National Clinical Research Center for Respiratory Disease, Guangzhou Institute of Respiratory Health, The First Affiliated Hospital of Guangzhou Medical University, Guangzhou 510120, China

**Keywords:** infectious disease, seasonal human coronavirus, spike antigen, T-cell epitope, cross-reactivity, virus-like particle vaccine, mRNA vaccine, pan-coronavirus vaccine

## Abstract

Seasonal human coronaviruses (HCoVs), including HCoV-229E, HCoV-NL63, HCoV-OC43, and HCoV-HKU1, circulate globally in an epidemic pattern and account for a substantial proportion of common cold cases, particularly in infants, the elderly, and immunocompromised individuals. Although clinical manifestations are typically mild, these HCoVs exhibit ongoing antigenic drift and have demonstrated the potential to cause severe diseases in certain populations, underscoring the importance of developing targeted and broad-spectrum vaccines. This review systematically examines the pathogenesis, epidemiology, genomic architecture, and major antigenic determinants of seasonal HCoVs, highlighting key differences in receptor usage and the roles of structural proteins in modulating viral tropism and host immunity. We summarize recent advances across various vaccine platforms, including inactivated, DNA, mRNA, subunit, viral-vectored, and virus-like particle (VLP) approaches, in the development of seasonal HCoV vaccines. We specifically summarize preclinical and clinical findings demonstrating variable cross-reactivity between SARS-CoV-2 and seasonal HCoV vaccines. Evidence indicates that cross-reactive humoral and cellular immune responses following SARS-CoV-2 infection or vaccination predominantly target conserved epitopes of structural proteins, supporting strategies that incorporate conserved regions to achieve broad-spectrum protection. Finally, we discuss current challenges in pathogenesis research and vaccine development for seasonal HCoVs. We propose future directions for the development of innovative pan-coronavirus vaccines that integrate both humoral and cellular antigens, aiming to protect vulnerable populations and mitigate future zoonotic spillover threats.

## 1. Introduction

Since the 1960s, seven human coronaviruses (HCoVs) have been identified as circulating in human populations, which include HCoV-229E, HCoV-OC43, HCoV-NL63, HCoV-HKU1, severe acute respiratory syndrome coronavirus (SARS-CoV), Middle East respiratory syndrome coronavirus (MERS-CoV), and SARS-CoV-2 [1,2,3,4,5,6,7]. HCoV-229E, HCoV-OC43, HCoV-NL63, and HCoV-HKU1 typically cause mild symptoms such as nasal congestion, rhinorrhea, sore or scratchy throat, cough, sneezing, and low-grade fever in epidemic patterns [8]. Therefore, these four low-pathogenicity coronaviruses are commonly referred to as seasonal HCoVs or common cold-associated coronaviruses. The other three HCoVs, SARS-CoV, MERS-CoV, and SARS-CoV-2, are associated with a high incidence of severe respiratory diseases, including acute respiratory distress syndrome (ARDS), pneumonia, various viral respiratory syndromes, and severe bronchiolitis [9]. Fortunately, no new cases of severe acute respiratory syndrome (SARS), which is caused by SARS-CoV, have been reported since the outbreak subsided in 2002–2004. However, MERS-CoV continues to cause sporadic infections with limited human-to-human transmission, whereas the newly emerged SARS-CoV-2, the causative agent of coronavirus disease 2019 (COVID-19), has remained prevalent in human populations with high transmissibility [10,11,12]. Although 60 years have passed since the discovery of the first HCoV, no vaccine has been approved for the prevention of these coronaviruses, with the exception of SARS-CoV-2 [13]. Specifically, among the top 50 wildlife viruses with zoonotic spillover potential, 21 viruses belong to coronaviruses [14]. Bats serve as one of the biggest reservoirs for coronaviruses. Evidence indicates that over 43% of collected bat samples are coronavirus-positive, harboring more than 4000 distinct coronaviral sequences [15]. Due to the high interspecies transmissibility and widespread prevalence of coronaviruses, developing pan-coronavirus vaccines against currently circulating strains as well as potentially future “X” coronaviruses remains urgently needed. Seasonal HCoV vaccines could provide a “platform” for rapid response to novel coronaviruses.

Vaccines function by introducing pathogen-derived agents into human bodies to activate protective immune responses [16]. The initial revolutions in vaccinology are exemplified by Louis Pasteur’s work on chicken cholera and rabies vaccines [17,18,19,20]. The parallel development of live-attenuated and inactivated vaccine platforms underpins the first major era of vaccine development and public health impact [21]. In the 20th century, scientists found that isolated pathogenic components could also confer protection against pathogens, leading to the development of corresponding vaccines for diphtheria, tetanus, *Haemophilus influenzae*, *Streptococcus pneumoniae*, and typhoid fever [22,23,24]. With the advancement of recombinant DNA technology in the late 20th century, recombinant hepatitis B virus (HBV) surface antigens (HBsAgs) were successfully produced via *Escherichia coli* and *Saccharomyces cerevisiae*, enabling the development of recombinant HBV vaccines, which provided protection against authentic viral challenge [25,26,27,28]. In 2000, Rino Rappuoli introduced the concept of “reverse vaccinology”, which enabled the identification of novel antigens utilizing genetic information [29]. Building on this approach, a universal five-antigen vaccine targeting *Neisseria meningitidis* serogroup B (MenB) was successfully developed and approved, demonstrating protection against 94% of 85 distinct MenB strains [30]. In recent years, protein structure-based antigen design has propelled vaccinology into a new era [31]. By integrating structural vaccinology with computational biology to enable rational antigen design, Rino Rappuoli and colleagues have termed this approach “reverse vaccinology 2.0” [32,33]. Based on this concept, scientists have successfully developed stabilized respiratory syncytial virus (RSV) Fusion (F) glycoprotein vaccines and SARS-CoV-2 prefusion Spike (S) vaccines [34,35,36].

Although both HCoV-NL63 and HCoV-HKU1 were initially isolated from patients with bronchiolitis or pneumonia, subsequent surveillance studies have shown that all four seasonal HCoVs commonly cause mild respiratory symptoms in winter, characterized by rhinorrhea, sore throat, cough, and low-grade fever. Severe disease manifestations are predominantly observed in infants, young children, the elderly, and immunocompromised individuals. As a result, seasonal HCoVs have historically received limited attention in the development of antiviral drugs and vaccines [37]. With the outbreak of COVID-19, substantial resources have been devoted to investigating the pathogenesis and developing therapeutic interventions against highly pathogenic SARS-CoV-2 infection [5]. Unlike seasonal HCoVs, SARS-CoV-2 exhibits higher transmissibility and a higher fatality rate [38,39]. SARS-CoV-2 causes a wide range of clinical symptoms, from asymptomatic infection to severe respiratory illness. Common symptoms overlap with those of seasonal HCoVs, such as fever, cough, headache, sore throat, diarrhea, and rhinorrhea. The severity of illness is influenced by multiple factors, including age, vaccination status, prior infection, viral variants, and underlying conditions, and may progress to hypoxemia, pneumonia, ARDS, sepsis and septic shock, acute kidney injury, cardiac complications, and thromboembolic events [40]. Another highly pathogenic coronavirus is MERS-CoV, which has caused sporadic infections and localized outbreaks since its emergence [7]. The clinical manifestations of MERS-CoV infection range from asymptomatic or mild respiratory symptoms to severe respiratory diseases and fatal outcomes [41]. Common symptoms include fever, cough, shortness of breath, and diarrhea. Pneumonia occurs frequently, but it does not always progress to this condition. The third highly pathogenic HCoV is SARS-CoV, which is no longer prevalent in humans. The typical symptoms of SARS include fever, chills, myalgia, headache, diarrhea, dry cough, and shortness of breath [42]. Most patients with SARS eventually developed pneumonia, requiring ventilator support for breathing. Due to the ongoing prevalence of SARS-CoV-2, MERS-CoV, and seasonal HCoVs, the development of broad-spectrum vaccines targeting these HCoVs and their variants is crucial to prevent future pandemics.

The outbreak of COVID-19 has profoundly reshaped our understanding of vaccinology [43]. Vaccines based on diverse technological platforms have been developed and authorized for the prevention of SARS-CoV-2 infection [44]. Unfortunately, no vaccine has yet been approved for combating seasonal HCoVs. Although seasonal HCoVs typically cause mild symptoms in most adults, these coronaviruses can still exhibit significant pathogenicity in infants and the elderly [45]. Moreover, these viruses remain persistently prevalent globally, accounting for 15–30% of all cases of the common cold [8]. Developing broad-spectrum vaccines targeting seasonal HCoVs is still essential to provide protection for susceptible populations and enhance preparedness for future pandemics. The Center for Infectious Disease Research and Policy (CIDRAP) at the University of Minnesota has launched a 6-year Coronavirus Vaccines Research and Development (R&D) Roadmap (CVR), aiming to develop “universal” coronavirus vaccines. The CVR provides a framework to guide funding and research initiatives that promote the development of such vaccines [13]. The majority of coronavirus vaccine-related reviews mainly focus on highly pathogenic HCoVs, including SARS-CoV-2 and MERS-CoV, while in this review, we specifically address the development of vaccines for seasonal HCoVs, despite the limited number of reported candidates. We begin by examining the pathogenesis and epidemiology of seasonal HCoVs, along with their genomic architectures and major antigens. We then comprehensively summarize recent advances in vaccine development targeting seasonal HCoVs and other respiratory viruses, with particular emphasis on the cross-reactivity observed between SARS-CoV-2 and seasonal HCoV vaccines. The literature search strategy encompassed databases from PubMed, Web of Science, Google Scholar, and preprint servers. Search keywords included “seasonal coronavirus”, “common cold coronavirus”, “HCoV vaccine”, and “coronavirus vaccine”. The search covered publications dated from 1960 to 2025. In the end, we discuss current challenges and propose future directions in developing pan-coronavirus vaccines against seasonal HCoVs and potentially all seven known HCoVs.

## 2. Pathogenesis and Epidemiology of Seasonal HCoVs

### 2.1. HCoV-229E

HCoV-229E is the first human coronavirus discovered in the 20th century. In 1966, Dorothy Hamre and John J. Procknow from the University of Chicago reported the isolation of this virus from specimens collected during the winter of 1962 [1]. They successfully recovered 5 infectious viruses from samples obtained from 4 individuals with upper respiratory infections and one asymptomatic individual. The virus isolated from specimen 229E was subsequently propagated in WI-38 lung fibroblast cell lines. Many cases of HCoV-229E infection can be asymptomatic or subclinical. The virus typically causes mild to moderate upper respiratory tract illness, with disease severity varying by age, immune status, and underlying medical conditions. Symptoms of HCoV-229E infection are associated with common cold, characterized by rhinorrhea, nasal congestion, sore throat, cough, sneezing, malaise, headache, low-grade fever, and myalgia (Table 1) [46]. Evidence indicates that HCoV-229E infections still can lead to pneumonia or acute respiratory distress syndrome (ARDS) in infants, the elderly, individuals with chronic diseases, and even in immunocompetent adults [47,48,49,50].

The genetic and phylogenetic evidence suggested that HCoV-229E originated from an ancestral alphacoronavirus circulating in bats, while camelids were implicated as a likely intermediate reservoir prior to human emergence [51,52]. Increased genetic drift and selective pressure have been observed in the S protein of HCoV-229E, with an evolutionary rate of 7.34 × 10^−4^ substitutions per site per year, contributing to rapid lineage diversification and enhanced intercontinental transmission [53]. HCoV-229E has been demonstrated to infect a wide range of cell lines, including WI-38, L-132, MRC-5, Huh7, and Mv1Lu [54,55,56,57,58]. The human aminopeptidase N (hAPN) is the primary receptor of HCoV-229E, mediating viral entry through interaction with the viral S protein [94,95]. Therefore, scientists have also established hAPN-overexpressing cell lines and animal models for subsequent drug and vaccine evaluations [59]. Interestingly, another report shows that 229E-related coronaviruses in camelids (dromedary camels) can utilize hAPN as the receptor to infect human cells, supporting the potential of camelids as reservoirs for 229E-like viruses with zoonotic transmission potential [96].

### 2.2. HCoV-OC43

HCoV-OC43 viruses were identified one year after the discovery of HCoV-229E (Table 1). In contrast to the standard tissue-culture methods used to isolate HCoV-229E, HCoV-OC43 was recovered from patients with upper respiratory tract diseases using human embryonic tracheal organ-culture (OC) techniques [60]. Subsequently, HCoV-OC43 was adapted to suckling mice and consistently produced encephalitic syndromes [2]. Like HCoV-229E, HCoV-OC43 typically causes mild upper respiratory tract infections, with symptoms associated with the common cold [46]. The clinical manifestations of HCoV-OC43 are generally characterized by sore throats, whereas coryza is more commonly observed during HCoV-229E infection [61]. However, HCoV-OC43 has been more frequently associated with lower respiratory tract diseases and hospitalizations in infants, the elderly, and immunocompromised individuals than HCoV-229E. Historically, HCoV-OC43 has occasionally been reported in association with encephalitis or neurologic disorders in animals and rarely in humans. However, these associations remain infrequent and lack definitive causal evidence in most cases [97,98,99]. Rare cases have demonstrated that HCoV-OC43 infection can lead to viral encephalitis and pneumonia in immunocompromised children [100,101,102]. Multiple prospective cohort studies have shown a strong association between HCoV-OC43 infection and respiratory complications, including pneumonia, tracheobronchitis, and bronchitis, as well as increased risk of mortality in elderly populations [103,104,105,106]. Additionally, evidence indicates that HCoV-OC43 infection can also cause pneumonia in immunocompetent adults [107].

Phylogenetic and molecular clock analyses have shown that HCoV-OC43 shares a common ancestor with bovine coronavirus (BCoV), both of which belong to betacoronaviruses [65,66]. HCoV-OC43 likely emerged through cross-species transmission from BCoV in the late 19th century, and might be associated with a respiratory disease pandemic reported during 1889–1890 [65]. Murine rodents are considered the likely ancestral reservoirs of HCoV-OC43, with subsequent host shifts occurring first into livestock as intermediate hosts and then into humans [67]. Multiple independent studies show that the S protein of HCoV-OC43 evolves at a rate of approximately 8 × 10^−4^ substitutions per site per year, with accelerated adaptive evolution concentrated in the S1 subunit, consistent with ongoing antigenic drift capable of facilitating recurrent reinfection in human populations [53,68,69]. HCoV-OC43 S protein specifically recognizes and binds to 9-*O*-acetylated sialic acids (9-*O*-Ac-Sia) present on host cell surface glycoconjugates, leading to the fusion of cellular and viral membranes and the subsequent release of viral genome into the host cell [64]. Due to the widespread presence of 9-*O*-Ac-Sia-conjugated glycoproteins on cellular membranes, HCoV-OC43 has been found to infect a broad range of cell types, including BS-C-1, fetal tonsil (FT), rhabdomyosarcoma (RD), Mv1Lu, TMPRSS2-expressing Vero E6, MRC-5, Huh7.5, HCT-8, HRT-18, and IGROV-1 cell lines [58,62,63,72,73,75]. HCoV-OC43 exhibits increased neurovirulence after passage in suckling mouse brain. Therefore, these murine-adapted strains have been extensively employed to assess the efficacy of antiviral drugs and vaccines in both BALB/c and C57BL/6 mouse models [70,71,74,97,98]. Tissue culture-adapted strains have also been utilized to establish infection models in wild-type mice; however, infected mice show either no clinical signs or delayed onset of pathological symptoms [70,97].

### 2.3. HCoV-NL63

In 2004, two groups in the Netherlands independently identified and characterized a novel HCoV, designated HCoV-NL or HCoV-NL63 (Table 1) [3,76]. Lia van der Hoek et al. isolated the virus from a nasopharyngeal aspirate sample obtained from a 7-month-old infant presenting with coryza, conjunctivitis, fever, and bronchiolitis in 2003, while Ron A. M. Fouchier and colleagues recovered this virus from a nose swab sample collected from an 8-month-old infant with pneumonia in 1988 [3,76]. Although both reports isolated HCoV-NL63 from infants with severe respiratory illness, the majority of infection cases present with mild upper respiratory symptoms resembling those of the common cold, including fever, rhinorrhea, dry cough, wheezing, respiratory distress, and coryza [77]. Prospective studies have demonstrated that 24% of HCoV-NL63-infected children develop croup, a condition characterized by hoarseness, a barking cough, and inspiratory stridor resulting from laryngeal obstruction [108]. High HCoV-NL63 viral loads are associated with severe lower respiratory tract symptoms, including bronchitis, bronchiolitis, and pneumonia [108]. The infection of HCoV-NL63 also causes severe respiratory illnesses in immunocompromised individuals and may result in complications in adults with co-infections [109,110,111,112].

Multiple reports have demonstrated that HCoV-NL63 likely originated from bat coronaviruses [81,82,83]. However, no confirmed intermediate host for HCoV-NL63 has yet been identified. Both HCoV-229E and HCoV-NL63 belong to alphacoronaviruses. Through phylogenetic analysis, Lia van der Hoek and colleagues indicated that HCoV-NL63 diverged from HCoV-229E in the 11th century [79]. Multiple recombination sites have been detected within mosaic genome structures of HCoV-NL63, providing evidence for interspecies recombination events [53,79]. Compared to HCoV-229E, HCoV-NL63 experiences comparatively weaker genetic drift and purifying selection, consistent with slower antigenic turnover and more prolonged co-circulation of distinct lineages. The estimated substitution rate in the *S* gene ranges from 3 × 10^−4^ to 6 × 10^−4^ substitutions per site per year, which is lower than that observed in HCoV-229E [53,79]. Although HCoV-NL63 (an alphacoronavirus) and SARS-CoV/SARS-CoV-2 (betacoronaviruses) are genetically distant, these viruses bind the same host receptor, human angiotensin-converting enzyme 2 (hACE2) [78]. Genetic and structural evidence indicate that this shared receptor usage is not due to homology between their S proteins or inheritance from a recent common ancestor capable of binding hACE2, but rather the result of independent convergent evolution [80,113]. Besides hACE2-overexpressing HEK293T cells, HCoV-NL63 has been found to infect LLC-MK2, Vero-B4, Caco-2 cell lines [3,80,84,85]. Recently, two studies have reported the successful establishment of HCoV-NL63 infection models utilizing K18-hACE2 transgenic C57BL/6 mice, as well as Ad5-hACE2-transduced *Ifnar1*^−/−^ C57BL/6 and BALB/c mice [59,86]. Notably, Donglan Liu et al. demonstrated that Ad5-hACE2-transduced wild-type mice did not support HCoV-NL63 infection, a finding that warrants careful consideration when evaluating the efficacy of vaccines using proper mouse models [59]. Additionally, it remains unclear whether pseudotyped virus neutralization assays can serve as partial substitutes for authentic coronavirus studies or be effectively applied in these animal models.

### 2.4. HCoV-HKU1

One year after the identification of HCoV-NL63, another novel coronavirus named HCoV-HKU1 was identified in the nasopharyngeal aspirate (NPA) sample from a 71-year-old patient with pneumonia (Table 1) [4]. Notably, retrospective studies have revealed that HCoV-HKU1 had also been present in an NPA sample collected from a 35-year-old pneumonia patient during the SARS-CoV outbreak in 2003. The NPA sample was SARS-CoV-negative, demonstrating a significant association between HCoV-HKU1 and pneumonia. HCoV-HKU1 infections have been associated with community-acquired pneumonia, including cases resulting in death [87,114]. Like other seasonal HCoVs, HCoV-HKU1 infections typically present with mild symptoms, such as fever, cough, sputum production, dyspnea, rhinorrhea, and sore throat. Less commonly, chills, rigors, myalgia, and headaches have also been reported. Notably, febrile seizures are frequently observed in pediatric patients infected with HCoV-HKU1 [87]. Pneumonia frequently occurs in elderly patients with underlying diseases, while acute bronchiolitis and asthmatic exacerbations are also occasionally observed in children. Another report shows that HCoV-HKU1 can also cause progressive pneumonia and even death in immunocompetent adults [115]. The rare co-infection with SARS-CoV-2 and HCoV-HKU1 may manifest with atypical clinical features and is associated with increased disease severity, prolonged hospitalization, bacterial superinfections, and cardiac complications, highlighting the need for multiplex diagnostic testing and careful administration of antiviral therapies in patients with preexisting cardiovascular conditions [116].

HCoV-HKU1 is phylogenetically closely related to murine coronaviruses among all known coronaviruses, providing strong support for a rodent origin [67,117]. To date, no direct bat or bovine progenitor has been identified. Phylogenetic and recombination analyses have revealed that recombination events in the S, accessory, and structural regions are prevalent among HCoV-HKU1 and other animal betacoronaviruses, with particularly frequent recombination observed between HCoV-HKU1 and murine hepatitis virus (MHV) [118]. Therefore, HCoV-HKU1 may originate from a major recombination event accompanied by multiple minor recombination events among betacoronaviruses. Among the four seasonal HCoVs, HCoV-HKU1 exhibits the slowest evolutionary rate in the *S* gene, with estimated substitution rates of 4.39 × 10^−5^ and 1.46 × 10^−4^ per site per year for the HKU1A and HKU1B clades, respectively [53]. These findings indicate weak genetic drift and limited selective pressure associated with prolonged regional persistence. Previously, HCoV-HKU1 has been shown to bind 9-*O*-Ac-Sia-conjugated glycoproteins to initiate infection, a characteristic shared with HCoV-OC43 [89]. Recent studies indicate that both cell surface sialoglycoconjugates and transmembrane protease serine 2 (TMPRSS2) function as receptors for HCoV-HKU1 [90,91]. The S protein of HCoV-HKU1 initially binds to sialoglycoconjugates, triggering a conformational change in the S trimer to an “open” state [92]. Subsequently, RBD of the “opened” S protein engages TMPRSS2, facilitating the fusion of viral and host cell membranes. Although the receptors of HCoV-HKU1 have been identified, culturing authentic HCoV-HKU1 viruses utilizing immobilized cell lines remains challenging. No wild-type mouse or transgenic mouse model exists for authentic HCoV-HKU1. Human ciliated airway epithelial cell cultures (HAE) and two-dimensional (2D) airway organoids have been successfully used to support HCoV-HKU1 replication. However, their complex and resource-intensive production procedures have substantially limited their widespread application in vaccine evaluation [88,93]. Recently, one study constructed a prototypic coronavirus by introducing target proteins of HCoV-HKU1 to the neurotropic strain of MHV (J2.2) [119]. The pathogenic role of the target proteins was clearly characterized in C57BL/6 mice, supporting the feasibility of reverse genetic systems. However, the chimeric MHV (J2.2) system is a surrogate model that allows study of individual HKU1 proteins but does not recapitulate natural infection. This represents a critical gap requiring urgent research investment, including the development of mouse-adapted strains via serial passage, which has been successfully applied to HCoV-OC43 and SARS-CoV-2.

### 2.5. General and Distinctive Features of Seasonal HCoVs

Beyond individual accounts of the four seasonal HCoVs, a more integrated and comparative perspective is essential to illuminate both their common virological features and distinct evolutionary dynamics. These seasonal HCoVs, including HCoV-229E, HCoV-NL63, HCoV-OC43, and HCoV-HKU1, belong to the genera Alphacoronavirus and Betacoronavirus and are enveloped viruses with positive-sense, single-stranded RNA genomes. They exhibit high mutation rates and a pronounced capacity for recombination, molecular traits that collectively enhance their genetic plasticity. This adaptability facilitates sustained transmission and recurrent resurgences in human populations. Notably, HCoV-229E and HCoV-OC43 have generated multiple variants displaying complex patterns of global dissemination and intercontinental spread, likely driven by ongoing antigenic drift and recombination in response to host immunity and environmental pressures [53]. Recent genomic surveillance studies have revealed substantial genetic differences between circulating modern seasonal HCoV strains and classical laboratory-adapted reference strains that have long been used in experimental virology and vaccine-related research [120,121]. This genetic divergence may undermine the translational relevance of in vitro findings, underscoring the need for updated reference systems that more accurately reflect currently circulating viruses. Furthermore, co-infection with seasonal HCoVs and other respiratory pathogens, such as influenza viruses, SARS-CoV-2, and rhinovirus C (RV-C), has been linked to increased clinical severity in certain populations, although the underlying mechanisms remain incompletely understood [116,122,123,124,125]. These interactions could involve synergistic modulation of the host immune response, competition for receptor binding, or altered cytokine dynamics, all of which may collectively influence disease progression and outcomes. Of particular concern is the potential for heterologous recombination between co-circulating coronaviruses, which could lead to the emergence of novel chimeric strains with altered pathogenicity or expanded host ranges [126,127]. Although most seasonal HCoVs are typically associated with mild upper respiratory tract infections, their significant evolutionary plasticity necessitates vigilant monitoring, as evidenced by previous zoonotic spillover events, such as those involving SARS-CoV, MERS-CoV, and SARS-CoV-2, which demonstrate the capacity of this viral family to generate pathogens of high public health consequence [128,129]. Consequently, integrating comparative genomics, molecular surveillance, and epidemiological data across seasonal HCoVs will be critical for anticipating evolutionary trajectories, improving the selection of laboratory reference strains, and informing the development of broad-spectrum medical countermeasures.

### 2.6. Public Health Impact and Rationale for the Development of Seasonal HCoV Vaccine

Due to the relatively mild symptoms associated with seasonal HCoV infections and the limited availability of comprehensive epidemiological data on their global endemicity, vaccine development against seasonal HCoVs has historically been deprioritized. This review provides a comprehensive summary of the virological characteristics, epidemiology, and current vaccine development status of seasonal HCoVs. Nevertheless, the development of virus-specific or broad-spectrum vaccines remains crucial for public health preparedness. Varicella-zoster virus (VZV), which causes chickenpox, typically results in mild disease in children but can lead to severe complications in adults and immunocompromised individuals [130]. The successful development and implementation of VZV vaccines have significantly reduced both disease burden and associated morbidity [131]. Similarly, although 95% of seasonal HCoV infections are mild, the remaining 5% in high-risk groups may represent millions of severe cases globally. Thus, developing seasonal HCoV vaccines is urgently needed for vulnerable populations. Moreover, vaccinating the general population against seasonal HCoVs could reduce transmission to those who cannot be vaccinated, potentially conferring herd immunity. Another significant data gap regarding seasonal HCoV infections is the potential for long-term sequelae, a phenomenon that has been observed in SARS-CoV-2 infections [132]. Whether seasonal HCoVs can lead to post-viral syndromes remains unknown. Post-infection follow-up studies are essential to address this question. Filling these knowledge gaps is critical for advancing efforts toward vaccine development, as it would help motivate funding agencies to prioritize research in this area, demonstrate to pharmaceutical companies the existence of a viable market opportunity, provide vaccine developers with clearly defined target product profiles, and assist policymakers in recognizing the importance of seasonal HCoV vaccines despite the typically mild nature of these infections.

## 3. Structures and Antigens of Seasonal HCoVs

### 3.1. Genomic Structures of Seasonal HCoVs

All four seasonal HCoVs share the same basic coronavirus genomic structures, with primary variations occurring in their accessory genes [133]. The genome sizes of HCoV-229E and HCoV-NL63 are approximately 27 kb, while those of HCoV-OC43 and HCoV-HKU1 are almost 30 kb. Each genome architecture contains a 5′ untranslated region (5′ UTR), followed by large replicase open-reading frames (ORFs) including ORF1a and ORF1ab, structural protein genes arranged in the order of Spike (S), Envelope (E), Membrane (M), Nucleocapsid (N), and a 3′ UTR. HCoV viral particles display “crown-like” surface projections observed by negative-stain or cryo-electron microscopy (Figure 1A) [134]. These characteristic structures arise from the densely packed array of S glycoproteins that project from the viral envelope. Additional envelope proteins, including E and M, are embedded within viral lipid bilayers and play crucial roles in organizing and stabilizing the envelope architecture, thereby ensuring proper spatial orientation of the S proteins. The N protein is one of the most abundantly expressed viral proteins during infection [135]. N binds the coronaviral positive-sense (+) genomic RNA and packages it into a highly compact ribonucleoprotein complex (RNP). N proteins also interact with M proteins at viral assembly sites located in the ER-Golgi intermediate compartment (ERGIC), driving the encapsidation of RNP into nascent virions [136]. The translated replicase polyproteins, including pp1a and pp1ab, are subsequently cleaved through proteolytic processing to generate 16 nonstructural proteins (NSPs) (Figure 1B) [137]. This proteolysis is catalyzed by two virus-encoded proteases, including NSP3-associated papain-like protease (PLpro) and NSP5 main protease (Mpro) [138,139]. During viral replication, NSP12 functions as RNA-dependent RNA polymerase (RdRp), catalyzing both the replication of the viral genomic RNA and the transcription of subgenomic RNAs [140]. NSP13 possesses both RNA helicase and NTPase activity, which unwinds RNA secondary structures and hydrolyzes NTPs to fuel translocation and unwinding [141]. To improve replication fidelity, NSP14 of seasonal HCoVs exhibits 3′-5′ exoribonuclease (ExoN) activity that enables proofreading of newly synthesized viral RNAs [142]. Conversely, NSP15 possesses uridylate-specific endoribonuclease (EndoU) activity [143]. NSP16 is a 2′-O-methyltransferase (2′-O-MTase) encoded at the 3′ end of ORF1ab [144]. Together with NSP10, it converts a Cap-0 structure into Cap-1 (m7GpppNm) by methylating the 2′ hydroxyl group of the ribose on the first transcribed nucleotide. The other NSPs have been shown to participate in viral replication and immune evasion through distinct mechanisms [145].

Most structural proteins and NSPs of four seasonal HCoVs are functionally conserved, despite variations in their sequence identity. Major differences in their genomic organization and function are primarily reflected in the receptor-binding specificity of S proteins, hemagglutinin-esterase (HE) proteins, and accessory proteins. HCoV-229E and HCoV-NL63 utilize hAPN and hACE2 as cellular receptors, respectively, while both HCoV-OC43 and HCoV-HKU1 enter host cells by binding to 9-*O*-Ac-Sia-conjugated glycoproteins. Recent studies have further demonstrated that TMPRSS2 serves as a functional receptor for HCoV-HKU1. Due to substantial differences in sequence and receptor usage, neutralizing antibodies are predominantly virus-specific, resulting in limited cross-protection among seasonal HCoVs provided by vaccines or antibodies. Specifically, HCoV-OC43 and HCoV-HKU1 encode an additional HE protein. The HE protein facilitates reversible attachment to sialylated receptors and collaborates with the S protein to regulate virion entry dynamics [146]. The esterase activity of HE may facilitate the release of progeny virions from decoy receptors, thereby influencing viral tissue tropism. Previous studies have shown that laboratory-adapted strains of HCoV-229E encode two accessory proteins, ORF4a and ORF4b, located between S and E (Figure 1A) [147]. ORF4a has been identified as a viroporin that facilitates viral production, a function analogous to that of the SARS-CoV ORF3a protein [148]. However, recent studies on clinical isolates suggest that the splitting of ORF4 into ORF4a and ORF4b may result from tissue-culture adaptation [121]. Clinical isolates of HCoV-229E retain the full-length ORF4 [149]. Phylogenetic and comparative analyses show that ORF3 in HCoV-NL63 is homologous to ORF4 of HCoV-229E. Functional studies reveal that ORF3 is an N-glycosylated protein that colocalizes with the E and M proteins within ERGIC [150]. Notably, ORF3 proteins are also incorporated into virions, suggesting a potential role as an additional structural protein of HCoV-NL63. Apart from the HE protein, HCoV-OC43 encodes three other accessory proteins. Among them, NS12.9 is a functionally homologous accessory protein to ORF4a of HCoV-229E, despite differences in their sequences. Similarly, this 12.9 kDa protein is located between S and E proteins, and functions as a viroporin that promotes efficient virion morphogenesis [151]. Another accessory protein encoded by HCoV-OC43 is NS2, which is located between NSP16 and HE proteins. Although the function of NS2 protein remains unclear, it shares significant homology with the NS2 proteins of MHV and BCoV [152]. MHV/BCoV NS2 is a 2′,5′-phosphodiesterase (PDE) that degrades 2′-5′ oligoadenylates (2-5A), preventing RNase L activation and protecting viral RNA from degradation [153,154]. Therefore, HCoV-OC43 NS2 likely antagonizes the innate immune response by interfering with the OAS-RNase L pathway. Within the ORF of the *N* gene, HCoV-OC43 encodes a small internal (I) accessory protein [155]. Recent studies show that this protein does not significantly alter viral replication or modulate inflammatory or innate immune responses, the precise function of which remains to be fully elucidated [119]. The I protein of HCoV-HKU1, also known as the N2 protein, shares the overlapping-frame genomic architecture with HCoV-OC43 I protein but exhibits distinct functional properties [88]. Recent evidence indicates that HCoV-HKU1 I protein reduces both viral load and immunopathology in a central nervous system (CNS) mouse model, demonstrating increased viral fitness through balanced modulation of host immune responses [119]. Another accessory protein encoded by HCoV-HKU1 is ORF4, located between the *S* and *E* genes. The biological function of ORF4 is still unknown.

### 3.2. Major Antigens of Seasonal HCoVs

The major antigenic proteins of seasonal HCoVs encompass four structural proteins, including S, E, M, and N. In the case of HCoV-OC43 and HCoV-HKU1, an additional HE protein can also be antigenic. The S glycoprotein exists as a trimer that projects from the surface of virions (Figure 1C). Most coronaviruses display approximately 20 to 60 S trimers per virion [156,157]. Each S protein consists of two functional subunits, S1 and S2. The recognition of target cells is mediated by the receptor-binding domain (RBD) located within the S1 subunit, which triggers the conformational transition of trimeric S from its prefusion to postfusion state [158]. The subsequent fusion of viral and cellular membranes is mediated by the fusion peptide (FP), heptad repeat 1 (HR1), and heptad repeat 2 (HR2). Due to their receptor recognition properties, neutralizing antibodies (nAbs) targeting the S protein, particularly the RBD region, can effectively block the entry of coronaviruses. Most HCoV vaccines utilize the S or RBD as antigens, and several vaccines targeting SARS-CoV-2 S/RBD have already been approved [44]. To increase the probability of B-cell receptors (BCRs) recognition and promote B-cell clonal expansion directed at virus-neutralizing sites, researchers have employed diverse vaccine design strategies to properly present neutralizing epitopes. Virus-like particles (VLPs), such as self-assembling protein polymers, enveloped VLPs (eVLPs), and chimeric VLPs, mimic the structural features of authentic viruses and present native-like antigenic surfaces, while lacking the viral genome [159,160]. Owing to their nanometer-scale size, VLPs can efficiently drain into lymphoid tissues and be directly presented by follicular dendritic cells (FDCs) to B cells as intact nanoparticles. Another strategy is inspired by structural determination of the stabilized prefusion S trimer [34]. Most trimeric S proteins on virions maintain the prefusion conformation, which efficiently preserves neutralizing epitopes [156]. To increase the percentages of prefusion S trimers for subsequent structural determination and vaccine design, two adjacent proline (2P) substitutions are introduced into the central helix (CH)/HR1-connecting region of the S2 subunit [35,95,161]. Due to the rigid cyclic side chains of 2P residues, this modification stabilizes the S trimer in its antigenically relevant prefusion conformation. Another challenge encountered during the production of homogenous S antigens is the premature shedding of S1 subunits [162]. The full-length S protein is pre-cleaved at the S1/S2 site, resulting in the noncovalent association between S1 and S2. These S1 subunits may shed from S trimers and act as decoy antigens by sequestering nAbs. To avoid S1 shedding, mutations at S1/S2 cleavage sites are also commonly incorporated into vaccine design [163]. In addition to prefusion stabilization and cleavage site mutations, other structural-guided substitutions have been implemented to increase the expression yields and thermal stability of trimeric S antigens [35].

The full-length S proteins and RBD proteins are commonly utilized as primary antigens for eliciting humoral immunity. Other structural proteins, including E, M, N, and HE, as well as conserved S2 regions of S, are predominantly employed as T-cell antigens for inducing cellular immunity [164]. These antigens are less susceptible to nAbs but provide cross-reactive T-cell epitopes shared among seasonal HCoVs and highly pathogenic coronaviruses [165,166]. CD4^+^ and CD8^+^ T cells recognize these conserved peptides when presented by HLA class II and class I molecules, respectively, triggering cellular immune responses that contribute to the clearance of heterologous viral infections and establish immunological memory against potential re-infections. Studies have shown that certain HCoV-OC43 epitopes derived from S and N are cross-reactive with SARS-CoV-2 homologs [167]. The prior infection with HCoV-OC43 or immunization with conserved N epitopes provides protection against SARS-CoV-2 infection and reduces lung damage [168]. Similarly, HCoV-229E or HCoV-NL63 infection-induced cross-reactive T-cell responses contribute to the protection against SARS-CoV-2 infection, suggesting the presence of shared cross-reactive T-cell epitopes among both seasonal HCoVs and SARS-CoV-2 [59]. Conversely, the immunization with conserved S2 subunit of SARS-CoV-2 also induces cross-reactivity against seasonal HCoVs. We previously developed a ferritin-based nanoparticle vaccine targeting both RBD and HR of SARS-CoV-2 [160]. The HR-containing nanoparticle vaccine elicited higher titers of cross-neutralizing antibodies and more potent cross-reactive T-cell responses against HCoV-229E and HCoV-OC43 compared to the RBD-only nanoparticle vaccine. The HR region within the S2 subunit harbors numerous cross-reactive CD4^+^ and CD8^+^ T-cell epitopes. Therefore, HR-specific T-cell responses may enhance B-cell affinity maturation and broaden the overall antibody responses. Although the E, M, N, and S2 subunits of S proteins potentially induce high levels of cross-reactive T-cell responses due to their considerable sequence homology, vaccines targeting these antigens alone generally fail to induce nAbs against heterologous viral infections. Both humoral antigens and cellular antigens should be incorporated in the development of broad-spectrum seasonal HCoV vaccines to achieve comprehensive and durable protection.

## 4. Vaccine Types Targeting Infectious Diseases

### 4.1. Inactivated Vaccine

The inactivated vaccine consists of virus particles cultivated in culture and subsequently inactivated to eliminate their ability to replicate, while preserving the structural antigens necessary to elicit an immune response (Figure 2) [169]. These vaccines present the entire viral particle to the immune system, thereby stimulating a broad antibody response targeting multiple viral proteins, especially structural proteins presented on virions. Due to the inability of the inactivated pathogen to cause disease, such vaccines generally offer a high degree of safety. However, because the inactivated virus does not undergo replication, these vaccines often induce a weaker cellular immune response and typically require adjuvants, multiple doses, or booster injections to achieve and sustain robust, long-term immunity [170]. The production process involves precise virus cultivation, thorough validation of inactivation, and stringent quality control measures to ensure batch-to-batch consistency and overall safety [171].

Inactivated vaccines have been employed for centuries [172,173]. The advent of reliable inactivation techniques enabled the development of modern viral vaccines using chemical or heat-based inactivation methods. Early influenza vaccines consisted of whole inactivated viruses cultivated in embryonated eggs and inactivated using either formalin or β-propiolactone [174]. With the improvement of inactivation and splitting agents, inactivated trivalent (TIV) and quadrivalent (QIV) influenza vaccines are still widely used for viral prevention [175]. During the COVID-19 pandemic, several inactivated SARS-CoV-2 vaccines employed traditional inactivated whole-virus approaches, which involve culturing SARS-CoV-2 in cell lines and subsequently inactivating the virus using chemical agents such as β-propiolactone or formalin derivatives [176,177]. However, the inactivated vaccine strategy is not universally applicable to all pathogens. Early attempts to develop an inactivated RSV vaccine using formalin-inactivated whole virus caused enhanced respiratory disease (ERD) in vaccinated infants after natural RSV infection [178]. Similarly, the formalin-inactivated measles vaccine, SARS-CoV vaccine, and MERS-CoV vaccine have also failed to induce protective immune responses upon infection [179,180,181]. Therefore, inactivated vaccines targeting specific viruses require careful safety evaluation in preclinical animal studies.

### 4.2. DNA Vaccine

The DNA vaccine introduces genetically engineered plasmid DNA encoding one or more pathogen-derived antigens into host cells, where the cellular machinery transcribes and translates these genes to produce antigenic proteins in situ [182]. These endogenously synthesized antigens are subsequently processed and presented via major histocompatibility complex (MHC) molecules, thereby activating both humoral and cellular immune responses, including cytotoxic T lymphocyte responses that are often challenging to elicit with conventional inactivated vaccines [183]. DNA vaccines offer advantages such as high stability, ease of production and modification, rapid adaptability to emerging pathogenic strains, and a favorable safety profile due to the absence of replicating pathogens (Figure 2). However, their immunogenicity in humans has historically been lower compared to that in small animal models, which has led to the development of enhancement strategies, including optimized plasmid backbones, codon usage refinement, molecular adjuvants, electroporation-based delivery, and heterologous prime-boost regimens [184].

The concept of recombinant DNA technology in vaccine development emerged in the late 20th century. Researchers showed that direct injection of plasmid DNA encoding an antigen could induce antigen expression in host tissue and generate immune responses in animals. Heather L. Davis et al. showed that the DNA vaccine encoding HBsAg induced rapid and durable antigen-specific antibodies in mice, suggesting enhanced humoral immunity [185]. Concurrently, two independent groups revealed that DNA vaccines encoding influenza hemagglutinin (HA) glycoproteins and nucleoprotein (NP) protected mice from authentic viral challenge, demonstrating both HA-specific humoral immunity and NP-specific cytotoxic T lymphocyte responses [186,187]. Since 2000, hundreds of clinical trials of DNA vaccines have been conducted [188]. However, only one DNA vaccine targeting SARS-CoV-2, named ZyCoV-D, has received emergency use authorization (EUA) in India [189,190]. The ZyCoV-D vaccine elicited robust nAbs and Th1-biased cellular responses across multiple animal models and in humans. DNA vaccines represent a valuable and flexible platform, particularly suitable for settings requiring thermostability and simplified manufacturing processes. Future research should focus on enhancing their immunogenicity in humans to facilitate broader clinical application.

### 4.3. mRNA Vaccine

mRNA vaccines utilize a short, synthetic strand of messenger RNA (mRNA) encapsulated in lipid nanoparticles to deliver genetic instructions into human cells, enabling these cells to temporarily produce a harmless fragment of a pathogen, typically a surface protein antigen [191]. This antigen subsequently activates both the antibody-mediated and T-cell-mediated arms of the immune system, without exposing the recipient to live or inactivated pathogens. The technology allows for rapid design and manufacturing, as once the antigen sequence is identified, the corresponding mRNA can be quickly synthesized using a standardized production platform that can be readily adapted to new targets or variants, thereby offering platform versatility and scalable manufacturing (Figure 2) [192]. Key benefits of mRNA vaccines include robust immunogenicity demonstrated in clinical trials and real-world applications, absence of risk for vaccine-induced infection, and efficient sequence updates to address variant strains [193]. However, many mRNA vaccines require strict cold-chain storage and transportation to ensure stability, posing logistical challenges in low-resource settings [194]. Transient reactogenicity is common, including injection-site pain, fatigue, and fever. Although rare, serious adverse events, such as myocarditis observed in certain demographic groups following administration of some COVID-19 mRNA vaccines, have been reported and continue to be closely monitored through ongoing pharmacovigilance efforts [195]. Additionally, the requirement for specialized manufacturing processes and stringent quality control in lipid nanoparticle formulation increases initial complexity [196]. Overall, mRNA vaccines constitute a potent and versatile tool in modern vaccinology, offering rapid development and high efficacy, yet accompanied by logistical and communication challenges that must be effectively addressed to maximize public health impact.

The concept of mRNA vaccine originates from the successful in vivo expression of proteins utilizing in vitro transcribed (IVT) mRNA delivered into animal models [182,197]. Accompanied by the development of lipid nanoparticles, the first liposomal mRNA vaccine encoding influenza NP was demonstrated to induce virus-specific cytotoxic T lymphocyte responses in mice [198]. The development of mRNA vaccines for the prevention of infectious diseases has progressed rapidly since 2010. Numerous lipid nanoparticle-encapsulated mRNA vaccines targeting pathogens such as influenza virus, rabies virus, HIV-1, MERS-CoV, and Zika virus have been developed, with several advancing into clinical trials [199,200,201,202,203]. The pivotal breakthrough for mRNA vaccines occurred in late 2019 with the onset of the COVID-19 pandemic. By December 2020, two mRNA vaccines, named BNT162b2 and mRNA-1273, which were developed by Pfizer-BioNTech and Moderna, respectively, had received EUA in the United States [204,205]. To date, millions of individuals worldwide have been vaccinated with mRNA-based formulations, which have demonstrated high efficacy in preventing SARS-CoV-2 infection and significantly reducing disease severity.

### 4.4. Subunit Vaccine

A subunit vaccine is a form of immunization that incorporates only specific, purified components of a pathogen, such as proteins, peptides, or polysaccharides, rather than using whole live or inactivated organisms [206]. These antigens are particularly selected based on their ability to elicit robust and protective immune responses while eliminating the risks associated with exposure to the entire pathogen. Subunit vaccines can be developed through recombinant DNA technology, synthetic peptide synthesis, or by purifying native pathogen-derived components from cultured microbes [207]. Key advantages include an excellent safety profile, as they contain no live infectious agents and pose no risk of reversion to virulence, reduced reactogenicity compared to whole-cell or live-attenuated vaccines. Subunit vaccines are also capable of directing the immune response toward well-defined protective antigens, thereby enhancing specificity and minimizing non-specific or adverse immune reactions (Figure 2). Additionally, these vaccines are typically stable and compatible with large-scale production using standardized biochemical methods. However, subunit vaccines also have notable disadvantages. Due to their limited antigenic content, they are often inherently less immunogenic and typically require adjuvants and multiple booster doses to induce strong and durable immune responses [208,209]. These vaccines may elicit a more restricted immune profile, often skewing toward humoral immunity at the expense of cellular immunity, which could compromise their efficacy against pathogens that require robust T-cell-mediated defenses.

The earliest foundations of subunit vaccines can be traced back to the late 19th through mid-20th century. Researchers found that protective immunity could be achieved by exposure to components of pathogens. Margaret I. Brady and I. G. S. Furminger developed an influenza subunit vaccine utilizing purified HA and neuraminidase (NA) proteins. This vaccine demonstrated significantly lower pyrogenicity compared to whole-virus vaccines while maintaining robust antigenicity and protective efficacy in animal models [210,211]. During the 1980s, the recombinant DNA technology enabled the safer and more scalable production of recombinant protein subunit vaccines using heterologous expression systems, such as bacteria, yeast, and cell lines. With advancements in antigen design driven by bioinformatics, structural biology, and artificial intelligence (AI), rationally designed subunit vaccines have markedly enhanced immunogenicity through the potentiation of both innate and adaptive immune responses [29,31,33]. Numerous subunit vaccines targeting infectious diseases, such as RSV, SARS-CoV-2, influenza, HBV, human papillomavirus (HPV), and varicella-zoster virus (VZV), have been successfully developed and are now approved for clinical use [212,213,214,215,216,217,218]. Specifically, both S and RBD regions of SARS-CoV-2 have been utilized to design SARS-CoV-2 subunit vaccines, which have demonstrated high levels of effectiveness and safety [213,214,219].

### 4.5. Viral Vector Vaccine

Viral vector-based vaccines utilize a harmless virus, known as the vector, to deliver genetic material encoding one or more antigens from a pathogen into human cells [220]. This enables the host immune system to recognize and respond to the antigens without exposure to the actual disease-causing agent. The viral vector is genetically modified to prevent replication and pathogenicity, with adenoviruses, poxviruses, and influenza virus being commonly used examples [221,222,223]. Upon infection of host cells, the delivered antigen-encoding gene is expressed intracellularly, processed, and presented via MHC molecules, thereby activating both antibody-mediated humoral and T-cell-mediated cellular immune responses. Because intracellular antigen production mimics natural infection more closely than other vaccine platforms, this approach may lead to more robust and durable immunity. Key advantages of viral vector vaccines include their ability to induce strong cellular immune responses, as well as rapid development and manufacturing once the antigen sequence is identified (Figure 2). However, a significant limitation of viral-vectored vaccines is the preexisting immunity to the vector [224]. The immune system can mount responses against the viral vector itself in individuals previously exposed to the corresponding virus or vaccinated with the corresponding viral-vectored vaccine, potentially neutralizing it before it delivers its genetic payload, thereby reducing vaccine efficacy. Certain adenoviral vector-based COVID-19 vaccines have been associated with rare but serious adverse events, such as vaccine-associated thrombosis with thrombocytopenia, raising safety concerns and necessitating enhanced pharmacovigilance [225]. Furthermore, large-scale manufacturing and the consistent maintenance of vector potency pose significant technical challenges.

Viral-vectored vaccines emerged from pioneering recombinant DNA and gene therapy research conducted between the 1970s and 1990s, evolving into a versatile and engineered class of platforms that have consistently demonstrated their effectiveness in outbreak responses and large-scale immunization programs. Originally adapted from adenovirus and poxvirus gene delivery systems, viral vector technology advances significantly through breakthroughs in reverse genetics for negative-sense RNA viruses, enabling the use of vesicular stomatitis virus (VSV), rabies, and paramyxovirus backbones, alongside the development of attenuated and replication-defective vectors, such as ΔE1 adenoviruses and modified vaccinia virus Ankara (MVA) [224]. In recent years, adenoviral platforms, including ChAdOx1-, Ad26-, and Ad5-based vectors, have enabled rapid adaptation and are frequently used in heterologous regimens for COVID-19 vaccination [226,227,228]. These applications highlight the speed and scalability inherent in viral vector platform-based vaccine development, while also uncovering rare but serious safety concerns, such as vaccine-induced thrombotic thrombocytopenia [225]. Due to declining global demand, AstraZeneca discontinued its ChAdOx1-S-nCoV-19 vaccine worldwide in 2024, although the company stated that the decision was unrelated to the rare risk of vaccine-induced thrombotic thrombocytopenia. Concurrently, mucosal and intranasal vector strategies, which are based on Newcastle disease virus (NDV), parainfluenza virus (PIV), influenza virus, and intranasal Ad/VSV candidates, have advanced in targeting respiratory pathogens [220,229,230,231]. Additionally, multivalent antigen designs, as well as integrated heterologous regimens combining viral vectors with mRNA or protein platforms, are prioritized to enhance breadth, durability, and deployability for both outbreak responses and routine immunization programs [232,233].

### 4.6. Virus-like Particle Vaccine

Virus-like particle (VLP) vaccines are a type of subunit vaccine that utilizes self-assembling protein structures mimicking the native architecture of viruses, while lacking viral genetic material and, thus, being non-infectious [159]. By presenting viral surface antigens in their natural conformation and highly ordered repetitive array, VLPs effectively stimulate robust B-cell and T-cell immune responses without the risk of viral replication. Due to their structural similarity to authentic virions, VLPs are efficiently recognized by antigen-presenting cells and activate the complement system, thereby enhancing humoral immunity and facilitating the production of high-affinity neutralizing antibodies (Figure 2). VLP-based vaccines exhibit an excellent safety profile, attributable to their inability to replicate or revert to pathogenic forms. Their design flexibility enables the incorporation of heterologous epitopes or conjugation with immunomodulatory molecules to target specific immune pathways [234]. Additionally, VLPs can elicit both mucosal and systemic immune responses when appropriately formulated or delivered [235]. However, manufacturing processes of VLP vaccines are often more complex and costly compared to simpler subunit vaccines, particularly when mammalian cell expression systems are employed [236]. Their development depends on precise protein folding, appropriate post-translational modifications, and accurate self-assembly. Certain VLPs exhibit inherent instability, necessitating cold-chain storage or the implementation of stabilization strategies [237]. Moreover, immune responses may be preferentially directed toward the carrier scaffold rather than the target epitope, potentially compromising the efficacy of displayed heterologous antigens [238].

Based on the presence or absence of lipid envelopes, VLP vaccines can be classified into enveloped VLP (eVLP) and non-enveloped VLP vaccines. With the discovery of diverse self-assembling protein nanoparticles and advances in chemical conjugation techniques, VLPs have emerged as highly effective scaffolds for antigen display. Commonly used self-assembling VLPs include the 24-mer ferritin, 60-mer lumazine synthase (LS), 60-mer bacterial E2p, as well as computationally designed 60-mer mi3 and I53-50. These VLPs have been extensively utilized in the development of vaccines against influenza, RSV, HIV-1, and HCoVs [159]. Apart from these protein VLPs, lipid-containing eVLPs are also utilized to display full-length membrane glycoproteins. Both plant- and mammalian cell-derived eVLP vaccines have been developed against influenza and SARS-CoV-2 [239,240,241]. These eVLP vaccines present glycoproteins embedded within a lipid bilayer, maintaining their native transmembrane orientation, quaternary structure, and membrane-proximal epitopes. By mimicking the morphological features of authentic viruses, they enhance the potential to elicit broadly nAbs that are specific to native conformational epitopes. VLPs are also promising platforms for developing broad-spectrum vaccines due to their capacity to display multivalent antigens. One study developed a mosaic-8 RBD VLP vaccine by co-displaying RBD antigens from eight different sarbecoviruses on mi3 VLPs [242]. This mosaic VLP vaccine elicited broadly protective immunity against both SARS-CoV-2 variants and antigenically mismatched zoonotic sarbecoviruses, demonstrating its potential for pan-sarbecovirus protection.

## 5. The Development of Vaccines Against Seasonal HCoVs

Although various types of vaccines have been developed against human infectious diseases, no vaccine has yet been approved for the prevention of seasonal HCoVs. Most licensed vaccines target influenza viruses, RSV, and SARS-CoV-2. We conducted a comprehensive review of preclinical vaccines directed against seasonal HCoVs and found that only a limited number of such vaccine candidates have been reported. Notably, many vaccines developed against the three highly pathogenic coronaviruses exhibit cross-reactivity to seasonal HCoVs, highlighting their potential as pan-coronavirus vaccines. In the subsequent sections, we systematically summarize recent advances in vaccine development for seasonal HCoVs, encompassing both target-specific and cross-reactive vaccines, as well as insights derived from natural infection.

### 5.1. Cross-Reactivity of Natural Infection and Inactivated Vaccines to Seasonal HCoVs

Previous studies have revealed that infections with HCoV-OC43 and HCoV-NL63 occur frequently in infants and are typically more common than those caused by HCoV-HKU1 and HCoV-229E [243,244]. However, the protective immune responses elicited by seasonal HCoVs are short-lasting, resulting in recurrent infections and frequent boosts in virus-specific antibody levels in adults [245]. Another two studies demonstrated that patients infected with SARS-CoV exhibited a significant increase in preexisting antibody titers against HCoV-OC43, HCoV-229E, and HCoV-NL63, suggesting the presence of cross-reactive epitopes among these HCoVs [246,247]. Similarly, during the COVID-19 pandemic, preexisting antibodies against S proteins of seasonal HCoVs were boosted by SARS-CoV-2 infection [248]. However, these antibodies primarily targeted the N protein and the S2 subunit of the S protein rather than the RBD protein [249,250]. These studies also showed that the preexisting antibodies against alphacoronaviruses, including HCoV-229E and HCoV-NL63, remained unchanged in SARS-CoV-2 convalescent individuals, indicating higher cross-reactivity between betacoronaviruses. The increase in S-specific antibodies was also observed for HCoV-HKU1 in SARS-CoV-2-infected patients [251]. Specifically, patients with higher levels of SARS-CoV-2 S-specific IgG, IgM, and IgA antibodies demonstrated enhanced cross-recognition to S proteins of both HCoV-OC43 and HCoV-HKU1. To precisely map the antibody landscape of sera from pre-pandemic and SARS-CoV-2 convalescent individuals, Andrew B. Ward and colleagues utilized both negative stain electron microscopy-based polyclonal epitope mapping (ns-EMPEM) and cryo-EMPEM to characterize these antibody specificities targeting S proteins of betacoronaviruses [252]. The results showed that antibodies against S2 subunit epitopes of HCoV-OC43 and HCoV-HKU1 were significantly increased in sera of SARS-CoV-2 convalescent individuals compared to those of pre-pandemic individuals. These epitopes, particularly a helix region spanning residues 1014 to 1030, are highly conserved across betacoronaviruses. The elevated levels of S2-reactive antibodies support the notion that SARS-CoV-2 infection can both boost preexisting seasonal HCoV-specific responses and elicit genuinely cross-reactive antibodies to conserved regions within the S2 domain.

The increased cross-reactivity to seasonal HCoVs following SARS-CoV-2 infection is predominantly directed to the N protein and the conserved S2 subunit of the S protein rather than S1 or RBD regions. Therefore, these cross-reactive antibodies may exhibit limited neutralizing potency but could still modulate disease severity. A retrospective study revealed that a recent documented seasonal HCoV infection, which was confirmed by comprehensive respiratory panel PCR (CRP-PCR) tests, was associated with less severe COVID-19 outcomes among hospitalized patients, despite comparable SARS-CoV-2 infection rates between seasonal HCoV-positive and seasonal HCoV-negative groups [253]. Another study utilized VirScan epitope profiling to probe antibody repertoires of COVID-19 patents and pre-COVID-19 era controls [254]. Consistently, the findings indicated that the prior exposure to seasonal HCoVs was associated with reduced disease severity in COVID-19. Cross-reactivity was observed between SARS-CoV-2 and seasonal HCoVs targeting specific linear epitopes, particularly within S2 regions of S proteins. Notably, the epitope spanning residues 811 to 830 of the S protein is conserved across SARS-CoV-2 and all four seasonal HCoVs, whereas the epitope encompassing residues 1144 to 1163 is shared specifically between SARS-CoV-2 and HCoV-OC43. These conserved epitopes suggest that preexisting antibodies against seasonal HCoVs targeting these regions may be reactivated or boosted upon SARS-CoV-2 infection, potentially modulating disease outcomes. Inconsistent with these findings, two additional longitudinal cohort studies revealed that the preexisting immune responses to S2 and N proteins of seasonal HCoVs were associated with increased susceptibility to SARS-CoV-2 infection and a higher risk of adverse outcomes in COVID-19 patients [122,255]. These discrepancies may arise from variations in the composition of COVID-19 patient cohorts and differences in the baseline health status of the individuals included in the studies. Notably, the prior immunization with any seasonal HCoV S proteins has been found to significantly impair the induction of nAbs against SARS-CoV-2 following subsequent SARS-CoV-2 S protein immunization in mice [255]. The pre-immunization with S proteins of HCoV-HKU1 or HCoV-NL63 specifically reduced the production of SARS-CoV-2 RBD-specific antibodies. These findings suggest that the preexisting immune responses to seasonal HCoVs may bias the germinal center response toward cross-reactive, non-neutralizing antibody specificities, thereby impairing the development of RBD-targeted nAbs.

Although infection with SARS-CoV-2 can boost antibodies targeting conserved regions, these antibodies are generally non-neutralizing and do not provide protection against seasonal HCoVs. However, many T-cell epitopes that provide cross-reactive T-cell immunity are highly conserved across seasonal HCoVs and SARS-CoV-2. By functionally characterizing the HLA class I and II predicted peptide pools of SARS-CoV-2, Alba Grifoni et al. revealed that SARS-CoV-2-reactive CD4^+^ T cells were detected in 40% to 60% of unexposed individuals, while SARS-CoV-2-reactive CD8^+^ T cells were found in a smaller proportion of such individuals, implying that CD4^+^ T-cell cross-reactivity was more prevalent than CD8^+^ T-cell cross-reactivity [256]. All SARS-CoV-2-unexposed individuals tested were seropositive for seasonal HCoV, including HCoV-OC43 and HCoV-NL63, consistent with prior exposure to common cold-associated coronaviruses. These cross-reactive CD4^+^ T-cell responses were relatively more focused on specific nonstructural proteins, such as NSP14, NSP4, and NSP6, and to a lesser extent against S proteins, suggesting that the inclusion of conserved non-S antigens in vaccine could effectively leverage preexisting immunity. Another study conducted genome-wide in silico prediction of SARS-CoV-2 CD4^+^ and CD8^+^ T-cell epitopes [257]. The study extracted and synthesized conserved epitopes shared between seasonal HCoVs and SARS-CoV-2. Many of these conserved epitopes elicited significant IFN-γ responses in peripheral blood mononuclear cells (PBMCs) from both COVID-19 patients and pre-COVID-19 individuals. Notably, these functional epitopes were predominantly derived from ORF1ab and the S2 subunit of S proteins, underscoring their potential as targets for pan-coronavirus vaccine development aimed at eliciting broadly cross-protective T-cell immunity.

Given the extensive cross-reactivity of humoral and cellular responses between seasonal HCoVs and SARS-CoV-2, inactivated SARS-CoV-2 vaccines may elicit protective immunity against seasonal HCoVs. Tanushree Dangi et al. reported that sera from both COVID-19 convalescent individuals and inactivated SARS-CoV-2 vaccine-immunized mice exhibited elevated levels of cross-reactive neutralizing antibodies against HCoV-OC43 [258]. The HCoV-OC43-specific IgG titers were 4,050 in pre-COVID-19 samples and 63,450 in COVID-19 samples, indicating a more than 15-fold increase in HCoV-OC43-binding antibodies following SARS-CoV-2 infection (Table 2). Mice immunized with γ-irradiated inactivated SARS-CoV-2 exhibited HCoV-OC43-specific IgG titers exceeding 10^3^, whereas unvaccinated controls showed undetectable levels. Furthermore, COVID-19 convalescent sera also protected mice from HCoV-OC43 infection. One study demonstrated that two doses of BBIBP-CorV inactivated SARS-CoV-2 vaccines significantly boosted preexisting IgG levels against S and S1 proteins of all four seasonal HCoVs in vaccinated individuals, with fold changes ranging from 5- to 500-fold depending on baseline antibody titers [259]. However, another study demonstrated that two doses of SinoVac inactivated SARS-CoV-2 vaccines induced an approximately 3-fold increase in preexisting IgG titers to S proteins of HCoV-NL63 but not HCoV-OC43, likely due to the high baseline levels of preexisting HCoV-OC43 S-specific antibodies prior to vaccination [260]. The elevated levels of preexisting immune responses may confer protective effects against SARS-CoV-2. Using immunoaffinity mass spectrometry (MS) quantification of MHC peptides eluted from HCoV-OC43-infected HEK293. CIITA cells, Lawrence J. Stern and colleagues identified that two MHC-II epitopes from S proteins, spanning residues 903 to 917 and 1085 to 1099, shared substantial homology with SARS-CoV-2 and potentially the other three seasonal HCoVs, demonstrating robust cross-reactive CD4^+^ T-cell recognition between HCoV-OC43 and SARS-CoV-2 homologs [167]. Another prospective healthcare workers (HCWs) cohort study showed that individuals with the highest baseline IgG levels against the N protein of HCoV-OC43 exhibited a substantially lower incidence of SARS-CoV-2 infection [261]. The result was compatible with short-lived cross-protection, likely mediated by T-cell immunity rather than cross-neutralizing anti-S antibodies in seasonal HCoVs. Consistently, a single prior intranasal HCoV-OC43 infection elicited polyfunctional CD8^+^ and CD4^+^ effector T cells that cross-reacted with SARS-CoV-2 peptides in HLA transgenic mice [168]. Both immunization with the N peptide (spanning residues 104 to 113) and infection with HCoV-OC43 reduced SARS-CoV-2 lung viral load and improved lung histopathology. Transgenic mice previously infected with HCoV-229E or HCoV-NL63 also showed reduced weight loss and more rapid SARS-CoV-2 clearance following challenge [59]. Passive transfer of sera from these infected mice failed to confer protection, whereas T-cell depletion reduced protective effects, supporting a T-cell-mediated cross-protection. Based on the cross-reactivity of natural infection and inactivated vaccines between seasonal HCoVs and SARS-CoV-2, the development of pan-coronavirus vaccines incorporating both humoral and cellular antigens may offer broad protection against both homologous and heterologous viral infections.

### 5.2. Nucleic Acid Vaccines for Seasonal HCoVs

Nucleic acid vaccines, including DNA and mRNA vaccines, have been widely utilized in the prevention of infectious diseases. Their rapid manufacturing procedure offers significant advantages in developing vaccines against emerging viruses and their subsequent variants. However, DNA vaccines are frequently overlooked due to their relatively low immunogenicity. To date, no DNA vaccine specifically targeting seasonal HCoVs has been reported. Only a few studies have explored the cross-reactivity of SARS-CoV-2 DNA vaccines with seasonal HCoVs. DNA vaccines against SARS-CoV-2, which include one approved ZyCoV-D vaccine, have demonstrated protective efficacy against authentic viral strains in both animal models and humans [189,190,262]. Dan H. Barouch and colleagues previously developed six DNA vaccine constructs encoding distinct forms of SARS-CoV-2 S proteins, including full-length S, cytoplasmic tail-truncated S.dCT, soluble ectodomain S.dTM, S1, RBD, and prefusion-stabilized S.dTM.PP [262]. All these vaccines induced antigen-specific antibodies and nAbs against both pseudotyped and authentic viruses in adult rhesus macaques, whereas the full-length S construct-immunized animals showed markedly reductions in viral titers after SARS-CoV-2 challenge. The Mesoscale Discovery (MSD) binding assays revealed that antibodies elicited by these DNA vaccines exhibited cross-binding activity against the full-length S proteins of four seasonal HCoVs [263]. These cross-reactive antibody responses showed a modest increase following the DNA vaccine boost and a significant enhancement after SARS-CoV-2 challenge (Table 2). However, these antibodies did not target RBD regions, likely due to the low sequence homology in these areas. More conserved S2 subunits could be the cross-reactive binding sites. Another study immunized HLA transgenic *Ifnar1*^−/−^ mice with three distinct DNA vaccines encoding SARS-CoV-2 S, M, and N proteins [168]. Through quantifying IFN-γ-producing and peptide-specific T cells in splenocytes and lung leukocytes from vaccinated mice, the authors identified several T-cell epitopes shared between SARS-CoV-2 and HCoV-OC43, indicating that the incorporation of conserved regions of structural proteins in DNA vaccines may elicit both humoral and cellular immune responses.

mRNA vaccines have demonstrated high efficacy against SARS-CoV-2 during the COVID-19 pandemic. Building on the success of the mRNA-1273 vaccine, Moderna has developed a preclinical-stage vaccine candidate, mRNA-1287, which encodes S antigens of all four seasonal HCoVs [264]. Recently, Tanushree Dangi et al. developed an mRNA vaccine encoding the stabilized S protein of HCoV-OC43 [265]. This vaccine elicited potent HCoV-OC43 S-specific antibodies with titers reaching 10^5^, and cross-reactive immune responses against MHV in mice. Notably, mice immunized with the mRNA vaccine were protected not only from homologous HCoV-OC43 challenge but also exhibited reduced disease severity and lower viral loads following heterologous challenge with MHV-A59. The passive transfer of sera from vaccinated mice conferred cross-protection. However, the depletion of CD4^+^ or CD8^+^ T cells failed to abrogate protection, indicating T-cell immune responses were unlikely to be the primary mechanism underlying this cross-protective effect. mRNA-1273 and BNT162b2 are two of the most widely administered mRNA vaccines globally [204,205]. However, their cross-reactivity with four seasonal HCoVs has been reported inconsistently. The mRNA-1273 vaccine has been shown to induce over 10^2^ titers of HCoV-OC43-reactive binding antibodies and cross-reactive CD8^+^ T cells in humans and animal models [258]. Complementary, two additional studies demonstrated that sera from mRNA-1273-vaccinated mice and humans effectively induced over 2- to 3-fold increase in cross-reactive antibodies against HCoV-NL63 and potently neutralized pseudotyped HCoV-NL63, but not HCoV-229E or HCoV-HKU1 [266,267]. However, the preexisting antibody responses against HCoV-NL63 were not boosted by mRNA-1273 vaccination, suggesting that the preexisting immunity to seasonal HCoV is unlikely to influence vaccine immunogenicity. In contrast, another study demonstrated that mRNA vaccination induced a back-boost in binding antibodies targeting S proteins of HCoV-OC43 and HCoV-HKU1, with a 2- to 4-fold increase compared to pre-vaccination, but did not elicit such responses for HCoV-229E or HCoV-NL63 [248]. This discrepancy may be attributed to differences in baseline levels of preexisting antibodies among individuals.

The BNT162b2 mRNA vaccine developed by Pfizer-BioNTech also demonstrated comparable cross-reactivity against seasonal HCoVs. Fatima Amanat et al. showed that sera from six individuals vaccinated with BNT162b2 exhibited elevated levels of cross-binding antibodies against S proteins from two seasonal betacoronaviruses, including HCoV-OC43 and HCoV-HKU1, with S-specific antibody titers exceeding 10^3^ [268]. The cross-binding activities to the alphacoronaviruses HCoV-229E and HCoV-NL63 were not significant, likely due to greater heterogeneity in preexisting immunity to these viruses. Another larger cohort study utilizing sera from 45 BNT162b2-vaccinated individuals showed that cross-reactive antibodies against S proteins of all four seasonal HCoVs increased approximately twofold after two-dose vaccination [269]. Similarly, a multicenter prospective HCWs cohort study revealed that one dose of BNT162b2 significantly induced a 2-fold increase in cross-reactive antibodies against S proteins of HCoV-OC43 and HCoV-HKU1, regardless of prior SARS-CoV-2 infection status [270]. Individuals with prior SARS-CoV-2 infection exhibited higher levels of cross-reactive antibodies compared to infection-naïve individuals after vaccination. However, no enhancement in cross-binding antibodies against HCoV-229E and HCoV-NL63 was observed in either infection-naïve or SARS-CoV-2-infected individuals following BNT162b2 vaccination. Recently, one study further investigated the neutralizing potency of these cross-reactive antibodies. Micaela Garziano et al. found that three doses of BNT162b2 vaccine did not significantly enhance nAbs against HCoV-OC43 [283]. However, both serum and saliva samples from individuals who were both infected with SARS-CoV-2 and vaccinated with an mRNA vaccine exhibited higher levels of nAbs against HCoV-OC43 compared to those from individuals receiving vaccination only. These findings indicate that the hybrid immunity induced by SARS-CoV-2 infection and mRNA vaccination may confer enhanced protective immune responses against HCoV-OC43 at both systemic and mucosal sites. Inconsistent with the above findings, two studies reported elevated levels of HCoV-NL63 S-reactive antibodies in sera from BNT162b2-vaccinated mice and humans [260,267]. However, these antibodies failed to neutralize authentic HCoV-NL63 viruses, suggesting that the conserved regions within S2 subunits of S proteins likely elicited these non-neutralizing antibody responses [267]. Further investigation of the cellular immune responses induced by these mRNA vaccines could provide valuable insights into their cross-reactivity against seasonal HCoVs.

### 5.3. Subunit Vaccines and Virus-like Particle Vaccines for Seasonal HCoVs

Both DNA and mRNA vaccines generate antigens through in vivo protein expression following administration into the human body, whereas subunit vaccines elicit immune responses by directly presenting purified antigens to immune cells. Given that the fundamental nature of virus-like particle (VLP) vaccines involves the direct presentation of recombinant subunit antigens, we comprehensively reviewed both subunit vaccines and VLP-based vaccines for seasonal HCoVs in this section. The development of subunit vaccines for HCoVs has been sparsely reported. Most studies primarily focus on cross-reactivity with SARS-CoV-2. Fatima Amanat et al. developed four subunit vaccines encoding S proteins of HCoV-229E, HCoV-NL63, HCoV-OC43, and HCoV-HKU1, respectively (Table 2) [271]. All these vaccines exhibited robust immunogenicity, eliciting homologous binding antibody titers ranging from 10^3^ to 10^4^. Notably, the S subunit vaccines for HCoV-229E and HCoV-NL63 also elicited cross-reactive antibodies against S proteins of both viruses at titers of 10^2^, indicating potential antigenic similarities between these two alphacoronaviruses. Similarly, the cross-reactivity of antibodies induced by S antigens of HCoV-OC43 and HCoV-HKU1 was also observed among betacoronaviruses, including SARS-CoV-2. However, mice immunized with seasonal HCoV S antigens were not protected against SARS-CoV-2 infection. The authors further demonstrated that the preexisting immune responses to these antigens did not compromise the protective efficacy of mRNA vaccines against SARS-CoV-2. Consistent with these findings, subunit vaccines targeting SARS-CoV-2 also generated cross-reactive antibodies to HCoV-OC43. Immunization with both the RBD and the S proteins of SARS-CoV-2 subunit vaccines induced approximately 10^3^ titers of HCoV-OC43-reactive antibodies [258]. Recently, Hiraku Sasaki et al. developed an intranasal subunit vaccine consisting of a conserved C-terminal fragment of the HCoV-OC43 S2 subunit fused to a nasal immuno-inducible sequence (NAIS) [272]. The intranasal administration of this vaccine induced systemic and mucosal immune responses against the S2 subunits of both HCoV-OC43 and SARS-CoV-2, with IgA and IgG antibody titers reaching 10^2^ and 10^3^, respectively. Sera from vaccinated mice demonstrated potent neutralizing activity against authentic HCoV-OC43, supporting further development of S2-based cross-coronavirus vaccines. Apart from the S2 subunit of S proteins, N proteins of seasonal HCoVs also harbor conserved T-cell epitopes shared with SARS-CoV-2. One study demonstrated that the immunization with a conserved N peptide vaccine spanning residues 104 to 113 protected mice from SARS-CoV-2 infection, although the protective efficacy against HCoV-OC43 challenge was not assessed [168]. The conservation of S2 and N proteins between seasonal HCoVs and SARS-CoV-2 not only facilitates cross-recognition of heterologous coronaviruses but also enhances T follicular helper (Tfh) cell-mediated cognate interactions with B cells. Building on this theoretical framework, one study developed a chimeric trimeric RBD (CTR) subunit vaccine by substituting the RBD region of HCoV-HKU1 S protein with that of SARS-CoV-2 [273]. The CTR vaccine maintained the RBD antigenic conformation and elicited potent nAb responses comparable to those induced by the wild-type S vaccine in vaccinated animals, while simultaneously recruiting HCoV-HKU1-derived Tfh cells to enhance germinal center responses.

The VLP vaccine represents a promising platform for displaying multivalent subunit antigens. Geoffrey B. Hutchinson et al. developed a pentavalent VLP vaccine by co-displaying S proteins of SARS-CoV, MERS-CoV, SARS-CoV-2, HCoV-OC43, and HCoV-HKU1 on the I53_dn5 nanoparticle scaffold [274]. The resulting mosaic_I53_dn5 vaccine elicited broad and cross-reactive antibody responses against all five betacoronaviruses, with antibody titers exceeding 10^5^, comparable to those induced by their corresponding monotypic I53_dn5 VLP vaccines (Table 2). Notably, the mosaic_I53_dn5 vaccine also induced cross-binding antibodies against S proteins of HCoV-229E, with titers reaching 10^3^, highlighting its potential as a pan-coronavirus vaccine. Another study computationally designed several universal antigens encompassing the optimized S 815-823 peptide and the stem helix epitopes [275]. Four distinct stem-helix epitope scaffolds (Ex) antigens were covalently conjugated to mi3 nanoparticles simultaneously, resulting in the assembly of an Ex_mosaic VLP vaccine. This vaccine elicited broad cross-reactive antibody responses against multiple betacoronaviruses in mice, with antibody titers exceeding 10^4^ for HCoV-OC43 and 10^2^ for HCoV-HKU1. Previously, Venezuelan equine encephalitis virus replicon particles (VRPs) have been successfully utilized to deliver S proteins of zoonotic SARS-CoV [276]. The VRP-S vaccine conferred protection against both epidemic and zoonotic strains of SARS-CoV in young mice. Similarly, Donglan Liu et al. developed two VRP vaccines encoding S proteins of HCoV-229E and HCoV-NL63, respectively [59]. The intranasal administration of these vaccines significantly reduced authentic viral replication in the lungs of transgenic *Ifnar1*^−/−^ mice. Apart from the above target-specific vaccines, many VLP vaccines targeting SARS-CoV or SARS-CoV-2 have demonstrated cross-reactivity to seasonal HCoVs. We previously developed a ferritin-based VLP vaccine displaying both RBD and HR antigens of SARS-CoV-2 [160]. The incorporation of HR antigens in VLP vaccine elicited potent nAbs against pseudotyped HCoV-229E and HCoV-OC43, achieving neutralization rates exceeding 15% and 40%, respectively. Compared with the RBD-only VLP vaccine, the HR-containing formulation induced stronger IFN-γ-producing T-cell responses against HCoV-OC43 but not HCoV-229E. These findings demonstrate that the inclusion of conserved S2 subunit antigens not only enhances broadly neutralizing activity but also promotes cross-reactive cellular immunity. Recently, a study developed two VLP vaccines utilizing the bacteriophage MS2 coat protein as the structural core [277]. These vaccines displayed S2 subunit antigens of SARS-CoV and SARS-CoV-2, respectively. By probing antigen-specific B cells from vaccinated mice, the authors demonstrated that B cells induced by the VLP-CoV-2 S2 vaccine exhibited broad cross-reactivity to HCoV-OC43, HCoV-HKU1, and HCoV-NL63, but not to HCoV-229E. In contrast, B cells obtained from VLP-CoV S2-vaccinated mice recognized only two seasonal HCoVs, including HCoV-OC43 and HCoV-HKU1. Another study examined the cross-reactivity against seasonal HCoVs utilizing sera from six cynomolgus macaques vaccinated with SARS-CoV-2 S protein-conjugated I53-50 VLP vaccines [269]. All six vaccinated macaques exhibited significantly increased titers of cross-reactive antibodies against S proteins of four seasonal HCoVs compared to baseline levels, with 16-, 6-, 106-, and 39-fold increase in S-specific IgG titers observed for HCoV-229E, HCoV-NL63, HCoV-OC43, and HCoV-HKU1, respectively. These findings suggest that the VLP vaccine may serve as an effective platform for the development of pan-coronavirus vaccines.

### 5.4. Viral Vector Vaccines for Seasonal HCoVs

Viral vector vaccines effectively elicit both humoral and cellular immune responses by mimicking the entry and protein expression patterns of pathogenic viruses, while employing harmless viruses as delivery vectors [220]. Among these viral-vectored platforms, adenovirus-based vaccines are commonly used for combating infectious diseases [221]. The ChAdOx1-S-nCoV-19 vaccine, which was developed by AstraZeneca and the University of Oxford for the prevention of SARS-CoV-2 infection, utilized a modified chimpanzee adenovirus as the vector to minimize potential preexisting immune responses against viral vectors [278]. In addition to eliciting strong immunogenicity against SARS-CoV-2, the vaccine has been shown to induce cross-reactive nAbs against HCoV-229E, but not against HCoV-NL63 or HCoV-HKU1 (Table 2) [266]. Administration of two doses of the ChAdOx1-S-nCoV-19 vaccine resulted in significantly higher nAb titers against pseudotyped HCoV-229E compared to a single dose. Recently, Maedeh Naghibosadat et al. developed a fowl adenovirus 9 (FAdV-9) vectored vaccine encoding SARS-CoV-2 S proteins (FAdV-9-S19) [279,280]. This vaccine elicited SARS-CoV-2-specific immune responses and protected mice against authentic SARS-CoV-2 challenge. Sera from vaccinated mice exhibited significantly higher nAb titers against HCoV-OC43, HCoV-NL63, and HCoV-229E compared to vector controls. Following intranasal challenge with HCoV-OC43 or HCoV-NL63, vaccinated K18-hACE2 mice showed markedly reduced oral viral shedding and lower infectious viral loads in the lungs. In cynomolgus macaques challenged with HCoV-229E, vaccination led to reduced viral shedding at multiple time points and decreased lung fibrosis scores upon necropsy. Moreover, vaccinated cynomolgus macaques mounted stronger Th1-biased cellular immune responses to both SARS-CoV-2 and HCoV-229E. These results demonstrated that the FAdV-9-S19 vaccine elicited robust humoral and cell-mediated cross-reactive immunity, providing cross-protection against seasonal HCoVs. Apart from zoonotic adenoviruses, human adenoviruses with low seroprevalence, such as the adenovirus serotype 26 (Ad26), are also commonly used to construct replication-incompetent vectors for vaccine development [284]. Based on the recombinant Ad26 vector, Johnson & Johnson developed the Ad26.COV2.S vaccine, which encoded the prefusion-stabilized SARS-CoV-2 S proteins [285]. The Ad26-vectored SARS-CoV-2 vaccines have exhibited robust immunogenicity and significant protective efficacy against authentic viral challenge in rhesus macaques [286]. Notably, Dan H. Barouch and colleagues demonstrated that sera from Ad26-S-vaccinated rhesus macaques showed enhanced cross-binding activities to S proteins of all four seasonal HCoVs, with S-specific antibody titers ranging from 10^2^ to 10^4^, compared to baseline levels [263]. Furthermore, subsequent SARS-CoV-2 challenge in these vaccinated animals led to an additional increase in cross-reactive immune responses, with antibody titers rising to exceed 10^3^ to 10^5^. Other viral vectors, including Ad5, VSV, and MVA, have also been employed in the development of coronavirus vaccines [258]. Tanushree Dangi et al. demonstrated that these vector-based vaccines, encoding S proteins of SARS-CoV or SARS-CoV-2, elicited cross-reactive antibody titers ranging from 10^2^ to 10^5^ against HCoV-OC43. However, the Ad5-vectored vaccine encoding SARS-CoV-2 S proteins failed to confer protection against HCoV-OC43 challenge. Notably, mice vaccinated with the Ad5-vectored vaccine encoding N proteins of SARS-CoV-2 exhibited significantly reduced viral loads in lungs following HCoV-OC43 challenge. These reports suggest that viral-vectored vaccines expressing conserved antigens can elicit both humoral and cellular immune responses and may, therefore, represent promising candidates for seasonal HCoV vaccines.

Due to the unique advantages of different vaccine platforms, scientists have proposed the administration of heterologous prime-boost regimen to enhance both humoral and cellular immune responses [287]. Previous studies have demonstrated that the heterologous prime-boost vaccination with sequential adenoviral-vectored ChAdOx1-S-nCoV-19 vaccine and BNT162b2 mRNA vaccine (ChAd-BNT) elicited significantly higher and broader nAb responses against multiple SARS-CoV-2 variants than homologous BNT-BNT vaccination [281]. Interestingly, Jan Lawrenz et al. revealed that sera from individuals receiving heterologous ChAd-BNT vaccines exhibited 1.5- to 4-fold increase in nAbs against authentic HCoV-OC43, HCoV-229E, and HCoV-NL63 compared to pre-vaccination levels (Table 2) [282]. These findings were largely corroborated by S protein-pseudotyped virus assays, indicating that the elicited nAbs primarily targeted the S proteins. Neutralizing potency to HCoV-HKU1 was not assessed in this study due to the limitation of susceptible cell lines for neutralization assays. Based on these findings, the heterologous vaccination may represent a promising strategy for eliciting robust cross-protective immune responses against infections caused by heterologous HCoVs.

## 6. Discussion

Since the 20th century, coronaviruses have continuously posed a significant threat to human health. Eco-epidemiological studies have demonstrated that numerous wildlife coronaviruses possess zoonotic spillover potential [14]. The development of virus-specific or broad-spectrum pan-coronavirus vaccines is, therefore, critical to prevent global transmission of currently circulating HCoVs and to prepare for future emerging coronavirus threats. With the outbreak of COVID-19, vaccines employing diverse technological platforms have been rapidly developed and authorized for the prevention of SARS-CoV-2 infection [44]. Although most vaccines based on the original viral strains have shown limited efficacy against SARS-CoV-2 variants, the widespread early vaccination has contributed significantly to population-level immunity, reducing disease severity [288]. Despite this progress in SARS-CoV-2 vaccine development, no vaccines have yet been approved for the prevention of infections caused by the other six HCoVs. Specifically, vaccines targeting four seasonal HCoVs, including HCoV-229E, HCoV-NL63, HCoV-OC43, and HCoV-HKU1, remain unavailable since their initial identification as early as 1966. Seasonal HCoV infections typically peak during winter and account for 15–30% of the common cold cases [8]. The clinical manifestations are generally mild, whereas their infections can still cause severe respiratory diseases in infants, the elderly, and immunocompromised individuals [45]. Therefore, developing vaccines against seasonal HCoVs is essential to protect susceptible populations.

### 6.1. Current Challenges in Seasonal HCoV Vaccines

In this review, we have comprehensively summarized the pathogenesis, epidemiology, major antigens, and both target-specific and cross-reactive vaccines associated with seasonal HCoVs. Several key challenges have been identified that significantly impede research into the pathogenesis of these viruses and the development of corresponding vaccines. Firstly, simplified tissue culture systems and animal models remain limited in their applicability to certain seasonal HCoVs. Among the four seasonal HCoVs, both cell line- and animal-based coronavirus infection models have been successfully established for HC-V-229E, HCoV-NL63, and HCoV-OC43, whereas developing such models for authentic and pseudotyped HCoV-HKU1 viruses is still challenging [59,73,84,98]. Without validated animal models demonstrating protection against authentic virus challenge, vaccine candidates face significant regulatory hurdles. The Food and Drug Administration (FDA) and European Medicines Agency (EMA) typically require challenge studies in relevant animal models before advancing to clinical trials for respiratory pathogens. The inability to assess protective efficacy means that immunogenicity data alone must guide development decisions, increasing the risk of clinical failure. Correlations of protection cannot be established without challenge models. Additionally, the absence of standardized models means each research group must invest many years developing and validating their own systems before vaccine testing can begin, fragmenting the field and delaying progress. Although human ciliated airway epithelial cells (HAE) and organoids support authentic HCoV-HKU1 infection, their resource-intensive production limits their large-scale evaluation in nAb studies [88,93]. Future studies should further validate the potential of organoid systems for specific endpoints, such as neutralization correlations and comprehensive protection assessments. No murine-based small animal models have been developed for HCoV-HKU1. Reverse genetic approaches using chimeric coronaviruses expressing HCoV-HKU1 proteins show promise but remain neither widely accessible nor standardized [119]. Alternatively, serial passage of HCoV-HKU1 in tissue culture and animal host to generate adapted strains could serve as a viable strategy for developing infection models, similar approaches of which have been successfully used to establish mouse-adapted models of HCoV-OC43, MERS-CoV, and SARS-CoV-2 [2,289,290].

Secondly, the heterogeneous receptor usage and tissue tropism among seasonal HCoVs complicate cross-viral comparisons and hinder the development of unified experimental models. Although all four seasonal HCoVs were originally isolated from respiratory specimens, they utilize distinct entry receptors, with hAPN for HCoV-229E, hACE2 for HCoV-NL63, 9-*O*-Ac-Sia for HCoV-OC43, and a combination of 9-*O*-Ac-Sia and TMPRSS2 for HCoV-HKU1 [64,78,89,90,94]. To date, robust cell culture systems with substantial levels of receptor expression have been established for all seasonal HCoVs except HCoV-HKU1 [58,63,85]. Moreover, these permissive cell lines exhibit limited overlap in cellular origin, making it challenging to propagate all four HCoVs in a single cell line. Therefore, engineering universal cell lines with broad spectrum by co-expressing multiple viral receptors may serve as a promising strategy for the simultaneous cultivation of the four seasonal HCoVs and the standardized evaluation of nAbs.

Thirdly, the determinants of severe disease and the functional roles of accessory proteins and HE proteins in seasonal HCoVs are poorly understood. Although seasonal HCoVs usually cause mild respiratory illnesses, they can lead to severe diseases in vulnerable populations such as infants, the elderly, and immunocompromised individuals [45]. The mechanisms underlying severe clinical manifestations, the neurotropism, and the contribution of accessory proteins to viral pathogenicity are still inadequately characterized. Many accessory proteins, such as ORF3, ORF4, NS2, NS12.9, I protein, and N2 protein, have unclear or only partially defined roles in immune evasion, tissue tropism, and pathogenesis [88,147,150,151,152,155]. Furthermore, the HE structural proteins of HCoV-OC43 and HCoV-HKU1 have been shown to facilitate viral entry by decoying virions from receptors, which may redirect virions to alternative attachment sites [146,291]. This functional property indicates that HE proteins may serve as potential antigens capable of eliciting nAbs and triggering antibody-dependent cellular cytotoxicity (ADCC), thereby contributing to protective immunity.

Fourthly, ongoing antigenic drift and recombination have been observed in seasonal HCoVs. Phylogenetic and molecular clock analyses indicate that the S proteins of four seasonal HCoVs accumulate substantial substitutions annually, driving rapid antigenic turnover and lineage diversification [53]. Most adaptive evolutionary mutations are concentrated in the S1 subunits, which are likely to contribute to recurrent reinfections in human populations. Notably, both HCoV-NL63 and HCoV-HKU1 have experienced interspecies recombination events, leading to the emergence of genetically distinct genotypes [79,114,118,292]. Continuous surveillance of the evolutionary dynamics of seasonal HCoVs is crucial for monitoring shifts in viral tropism and virulence, thus supporting more informed vaccine strain selection.

Fifthly, cross-reactive immune responses induced by heterologous coronaviral infections demonstrate limited breadth and cross-protective efficacy. Throughout this review, we have systematically analyzed the cross-reactivity between SARS-CoV-2 and seasonal HCoVs. However, cross-reactivity triggered by SARS-CoV-2 infection or vaccination predominantly targets conserved regions of the S2 subunit or N proteins and is often non-neutralizing [160,258,263,280]. In contrast, the nAb responses typically target the S1 or RBD regions of S proteins and are largely virus-specific [249,250,277]. Identifying and validating potent conserved neutralizing epitopes shared across seasonal and highly pathogenic HCoVs remains a significant challenge. Numerous conserved epitopes also elicit robust cross-reactive T-cell immune responses, enabling cross-recognition of heterologous coronaviruses and strengthening Tfh-cell-mediated cognate B cell interactions, which in turn enhance germinal center responses and promote the generation of nAb-secreting plasma cells [160,165,166].

Sixthly, preexisting immunity to seasonal HCoVs is common and heterogeneous across individuals. Due to recurrent seasonal infections caused by these common cold-associated HCoVs, preexisting immune responses are frequently observed in human populations [243,244]. Although evidence indicates that such preexisting immunity may not impair the efficacy of SARS-CoV-2 vaccines and can be back-boosted following SARS-CoV-2 infection or vaccination, it remains unclear whether these immune responses influence the effectiveness of vaccines against heterologous seasonal HCoV infections [248,260]. Notably, the back-boosted antibodies predominantly target conserved regions of seasonal HCoVs and exhibit limited neutralizing capacity, which may interfere with the development of virus-specific nAbs. Conflicting reports have shown that preexisting immune responses to seasonal HCoVs were associated with either increased disease severity in COVID-19 patients or altered susceptibility to SARS-CoV-2 infection in animal models, highlighting the necessity for caution when incorporating proper antigens that may elicit cross-reactive immune responses [59,168,253,255]. The antigenic sin, also known as immune imprinting, could be a unifying mechanism underlying the increased disease severity of COVID-19 in patients with preexisting immune responses to seasonal HCoVs [293,294,295]. Upon SARS-CoV-2 exposure or vaccination, preexisting memory B cells targeting conserved S2 or N epitopes of HCoVs may be preferentially recalled, leading to the rapid generation of cross-reactive but non-neutralizing antibodies, thereby competitively suppressing naïve B-cell responses against protective RBD epitopes, consequently resulting in the back-boost phenomenon without generating protective immunity against specific HCoVs. These conflicting reports also demonstrate the importance of balancing the quality and quantity of cross-reactive immune responses against seasonal HCoVs. The conflicting outcomes likely reflect differences in both beneficial and detrimental scenarios. The low-level preexisting CD4^+^ and CD8^+^ T-cell immunity targeting conserved epitopes may provide partial protection through viral clearance and enhanced germinal center responses. However, high-titer non-neutralizing antibodies may also saturate Fc receptors without effective viral neutralization, redirect immune responses away from protective RBD epitopes, impair de novo generation of SARS-CoV-2-specific neutralizing antibodies, and potentially form immune complexes contributing to immunopathology. Of note, the study design heterogeneity may also contribute to these contradictions. Retrospective versus prospective study designs differ in their ability to control for confounding factors. Key cohort characteristics, such as age distribution, comorbidities, and the timing and frequency of seasonal HCoV exposures, can influence study outcomes. The timing of immune assessment is also critical. Early infection stages, where T-cell immunity may be protective, contrast with chronic or severe disease phases, where non-neutralizing antibodies might contribute to immunopathology. Furthermore, differences in epitope specificity across studies, particularly those measuring S1 or RBD responses compared to those targeting S2 or N proteins, may explain divergent conclusions.

Lastly, protective immune responses to seasonal HCoVs are typically short-lived, resulting in frequent reinfections. Although seasonal HCoV-specific CD4^+^ T-cell responses and IgG humoral responses persisted over time, their magnitude did not correlate with the annual prevalence of seasonal HCoV infections [296]. Seasonal HCoV reinfections occur commonly, with cases detected as early as 6 months after initial infection and the majority occurring around 12 months post-infection [245]. These longitudinal cohort studies have shown that while immune responses to seasonal HCoVs are relatively durable, they do not confer complete protection against reinfection. Given the challenge of generating robust and long-lasting systemic and mucosal immunity, future vaccine development against seasonal HCoVs should focus on optimizing antigens, adjuvants, and delivery routes to achieve sustained protective immunity. Collectively, these challenges substantially hinder our understanding of seasonal HCoV pathogenesis and impede the development of broadly protective and durable vaccines. Addressing them requires coordinated advancements in model development, receptor engineering, rational design of conserved neutralizing antigens, surveillance of antigenic drift and recombination events, and refinement of vaccination platforms incorporating mucosal delivery strategies.

### 6.2. Strategic Framework for the Development of Seasonal HCoV Vaccines

Based on the current landscape and challenges in vaccine development for seasonal HCoVs, we propose strategic approaches to guide the design and implementation of broad-spectrum vaccines targeting seasonal HCoVs (Figure 3). Firstly, the design of universal antigen with T-B cell synergy for seasonal HCoVs is a priority to induce cross-protective immunity. Although developing a universal antigen across multiple HCoV genera remains challenging due to low protein sequence homology and diverse receptor usage, the advances in reverse vaccinology based on genetic information, structural vaccinology, and computational biology have significantly advanced the rational design of such antigens [33]. Structural and sequence design tools such as FoldX, ProteinMPNN, RFdiffusion, AlphaFold 3, and Rosetta enable backbone constrained sequence design, folding stabilization, and de novo antigen design, while tools like BepiPred, IEDB, and NetMHCpan facilitate the identification of B-cell and T-cell epitopes [297]. Leveraging these approaches, researchers have developed universal antigens for pathogens including MenB, HIV-1, influenza, and RSV [30,298,299,300]. An effective universal antigen formulation should deliberately integrate both humoral and cellular immune targets to induce broadly neutralizing antibodies (bNAbs) against validated conserved neutralizing sites, as well as robust and durable T cell responses capable of clearing infected cells and supporting germinal center reactions, including Tfh-cell-mediated affinity maturation. The S proteins of seasonal HCoVs contain both neutralizing B-cell epitopes and HLA-promiscuous T-cell epitopes, making them a predominant target for universal vaccine antigen design [301,302]. By integrating structural insights from antigen–antibody complexes with computational consensus sequence analyses, a rationally designed de novo antigen should incorporate conserved neutralizing B-cell epitopes and cross-reactive T-cell epitopes, while minimizing the exposure of strain-specific, immunodominant decoy epitopes that elicit narrow or non-protective immune responses. Typical targets include conserved conformational epitopes within RBD of the S1 subunit and cross-reactive HR regions in the S2 subunit. Conserved T-cell peptides derived from the N and ORF proteins can also be incorporated into antigen designs. Importantly, vaccine antigens must minimize exposure or immunodominance of conserved non-neutralizing epitopes that may misdirect B cell responses toward ineffective antibody production. Thus, rational antigen scaffold design and structural stabilization are essential strategies to focus immune responses on protective epitopes and finalize optimized universal vaccine antigens. Additionally, a sequential vaccination strategy, initiating with conserved epitopes to establish T-cell help, followed by administration of variable RBD epitopes, could serve as an alternative approach to mitigate immune focusing.

Secondly, multivalent antigen delivery can be utilized to enhance the efficiency and effectiveness of protective immunity (Figure 3). The primary goal of a broad-spectrum vaccine is to elicit diversified nAb or bNAb repertoires, as well as robust and broadly cross-reactive T-cell immune responses. Therefore, designed universal antigens or four distinct seasonal HCoV antigens should be properly and efficiently presented to the immune system in a way that maximizes simultaneous recognition, affinity maturation, and T-cell help. Optimized vaccine platforms are capable of co-delivering these antigens efficiently. mRNA and viral-vectored platforms offer flexible, sequence-based approaches for multivalent delivery by encoding multiple antigenic sequences within a single formulation or vaccination regimen. These platforms enable rapid updates to antigen composition in response to surveillance data, facilitate the co-expression of structurally distinct antigens, and can be engineered to modulate expression kinetics and cellular tropism to favor desired antigen processing pathways. Leveraging on these advantages, Moderna has developed a preclinical mRNA-1287 vaccine encoding S antigens of all four seasonal HCoVs [264]. In contrast, subunit vaccines and VLP platforms allow for precisely controlled, high-density multivalent presentation of conformational epitopes on particulate scaffolds. This configuration is particularly effective in cross-linking B-cell receptors and enhancing germinal center reactions, thereby driving the development of broad and high-affinity antibody responses. By turning particle geometry, antigen spacing, and valency, immune responses can be preferentially directed toward conserved neutralizing epitopes while reducing immunodominance of variable strain-specific regions. Given that seasonal HCoVs experience antigenic drift and exhibit substantial divergence in S protein sequences across both alphacoronaviruses and betacoronaviruses, multivalent strategies incorporating antigens from multiple circulating strains increase the likelihood of eliciting cross-reactive B-cell clones and broad T-cell immunity. A pentavalent I53_dn5-based VLP vaccine encoding S proteins of SARS-CoV, MERS-CoV, SARS-CoV-2, HCoV-OC43, and HCoV-HKU1 elicits broad and cross-reactive antibody responses to all five homologous coronaviruses as well as the heterologous HCoV-229E, supporting its potential as a broad-spectrum vaccine against seasonal HCoVs [274]. Notably, self-assembling protein-based VLPs are inherently immunogenic, potentially triggering non-specific immune responses. Therefore, selecting VLPs with lower immunogenicity or engineering them to minimize immunodominance is crucial for the development of safe VLP-based vaccines.

Thirdly, a heterologous prime-boost regimen via dual-route vaccination can be employed to induce robust cross-protective systemic and mucosal immune responses against infections caused by heterologous HCoVs (Figure 3). The clinical study utilizing a sequential adenoviral-vectored ChAdOx1-S-nCoV-19 vaccine followed by the BNT162b2 mRNA vaccine (ChAd-BNT) has demonstrated significantly higher and broader nAb responses against SARS-CoV-2 variants compared to homologous vaccination regimens [281]. Notably, this heterologous prime-boost strategy also exhibited enhanced cross-neutralizing activity against seasonal HCoVs [282]. Different vaccine platforms exhibit distinct immunological advantages, contributing uniquely to the overall immune response profile. DNA- and mRNA-based vaccines enable rapid and robust priming of antigen expression, which can be effectively boosted by subunit or VLP vaccine presenting stabilized prefusion S proteins or conserved epitope mosaics in particulate form, thereby directing B-cell affinity maturation with greater precision. Viral-vectored vaccines induce strong cellular immune responses and can be strategically combined with protein-based platforms to enhance the breadth of nAbs and strengthen T-cell immunity, while mitigating challenges such as anti-vector immunity. Furthermore, mRNA and viral vector platforms support iterative updates of antigens, preserving long-lasting cross-reactive immune memory originally primed by subunit or VLP components. As such, these integrative vaccination strategies synergistically maximize both the magnitude and quality of antiviral immunity. One major challenge in developing vaccines against respiratory infectious diseases is the difficulty of inducing potent and long-lasting mucosal immune responses. The “prime-and-pull” strategy has been proposed as a promising approach, in which systemic vaccination primes antigen-specific effector T cells, followed by localized administration of chemokines or immune modulators to “pull” these cells to the mucosal site of pathogen entry, thereby promoting the establishment of tissue-resident memory T cells [303,304]. Similarly, the prime-pull vaccination strategy using heterologous vaccine types has been shown to effectively leverage preexisting systemic immunity induced by a parenteral prime, enabling an intranasal vaccine booster to establish robust mucosal immune responses and immunological memory in the respiratory tract [305,306]. Mechanistically, the systemic prime generates circulating antigen-specific B and T cells, while the intranasal boost recruits and expands these cells, promoting their differentiation into local antigen-specific B cells that produce nAbs and effector T cells capable of eliminating infected cells. Because seasonal HCoVs infect the upper respiratory tract and mucosal immunity is key to preventing infection and onward transmission, we, therefore, propose that a heterologous prime-boost vaccination regimen, with the prime administered intramuscularly and the boost delivered intranasally, could represent a promising strategy to elicit robust systemic and mucosal immune responses against seasonal HCoVs. Given the unique advantages of various vaccine platforms, we recommend intramuscular priming using mRNA or subunit vaccines, followed by intranasal boosting with viral vector- or VLP-based vaccines. Combining systemic immunization, which induces high-affinity serum nAbs and systemic T cell responses, with mucosal vaccination, which elicits antigen-specific secretory IgA, tissue-resident memory T cells, and rapid frontline protection, offers a comprehensive and synergistic strategy for robust immune defense. Notably, the proposed heterologous prime-boost strategies cannot currently be validated for HCoV-HKU1, necessitating concurrent development of mouse models alongside antigen design.

## 7. Conclusions

Collectively, we have conducted a comprehensive analysis of the pathogenesis and vaccine development for seasonal HCoVs. Building on the current status and existing challenges in seasonal HCoV vaccines, we have also proposed a strategic framework for the design and implementation of broad-spectrum vaccines targeting these viruses. Although substantial knowledge has been accumulated on highly pathogenic HCoVs, our understanding of seasonal HCoVs remains limited. Designing more effective broad-spectrum vaccines against seasonal HCoVs requires a deeper elucidation of viral pathogenesis and antigenic determinants. Particularly, our understanding of infection-induced mucosal immunity is still inadequate. Further investigation into mucosal immunity will advance the development of broad-spectrum vaccines against seasonal HCoVs as well as the other respiratory viruses. The field will benefit from integrated translational pipelines that connect antigen discovery, standardized immunological assays, and iterative preclinical testing with clinical trials, complemented by global surveillance of HCoV antigenic drift and recombination. These coordinated efforts will enhance preparedness against seasonal HCoVs and provide critical guidance for pan-coronavirus vaccine development and broader respiratory virus control strategies.

## Figures and Tables

**Figure 1 vaccines-13-01168-f001:**
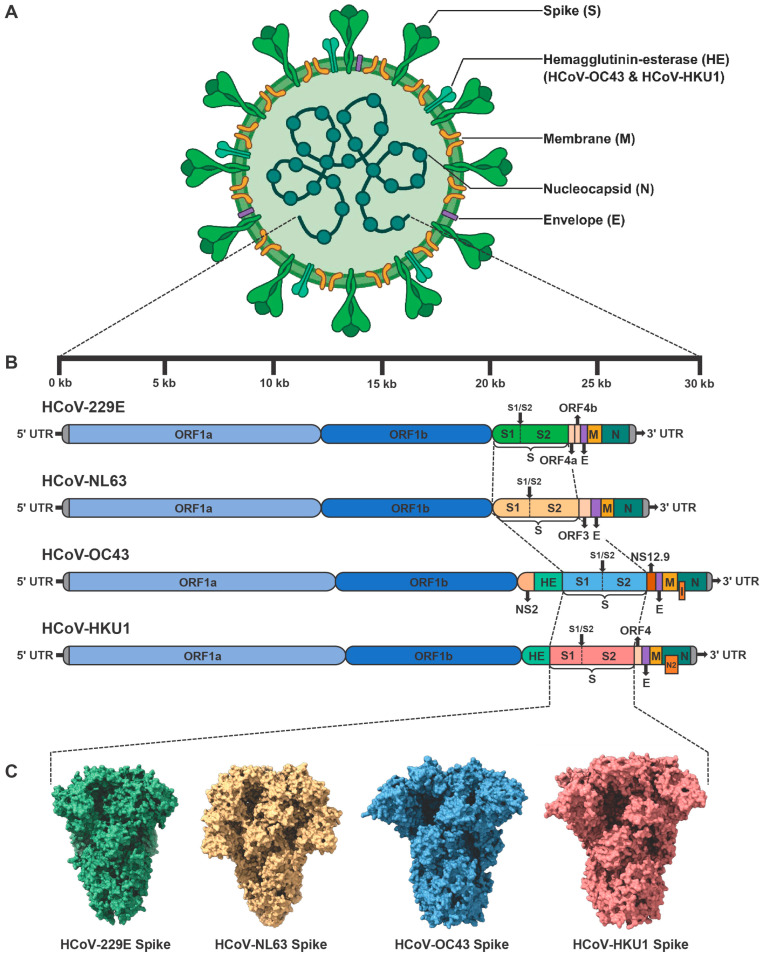
Genomic structures of seasonal HCoVs and electron microscopy-derived structures of their corresponding Spike (S) trimers. (**A**) Schematic representation of seasonal HCoV viral particles. (**B**) The genomic architectures of four seasonal HCoVs, including HCoV-229E, HCoV-NL63, HCoV-OC43, and HCoV-HKU1. Each viral genome comprises a 5′ untranslated region (5′ UTR), open-reading frame 1a (ORF1a), ORF1b, genes encoding structural proteins, accessory protein genes, and a 3′ UTR. S, Spike. E, Envelope. M, Membrane. N, Nucleocapsid. HE, Hemagglutinin-esterase. I, Internal protein. (**C**) Representative electron-microscopy structures of S trimers from four seasonal HCoVs, including HCoV-229E S trimer (PDB: 7CYC), HCoV-NL63 S trimer (PDB: 7KIP), HCoV-OC43 S trimer (PDB: 9BLK), and HCoV-HKU1 S trimer (PDB: 9BSW).

**Figure 2 vaccines-13-01168-f002:**
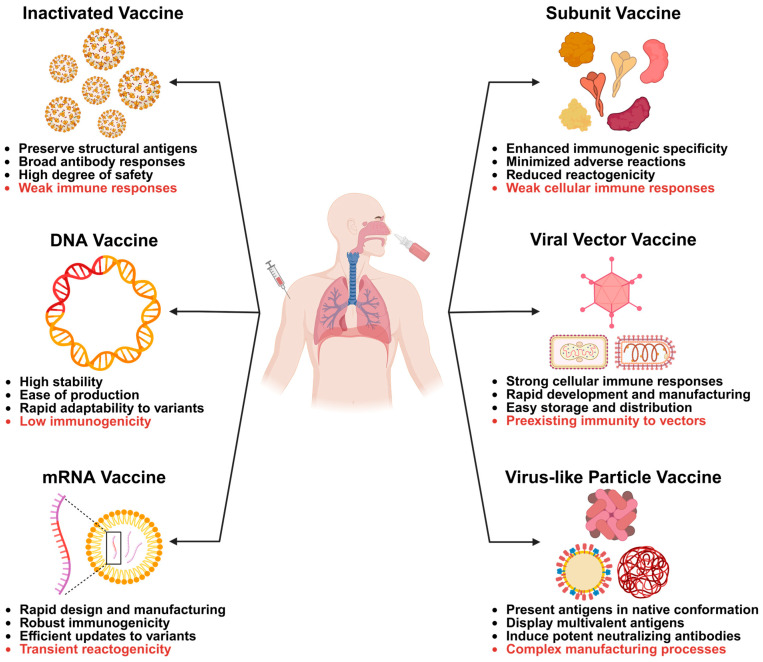
Major vaccine types targeting infectious diseases. Commonly used vaccines include inactivated vaccines, DNA vaccines, mRNA vaccines, subunit vaccines, viral vector vaccines, and virus-like particle (VLP) vaccines. These vaccines can be administered via intramuscular or intranasal routes. Intramuscular administration typically induces strong systemic immune responses, while the intranasal route specifically elicits potent mucosal immunity. Each vaccine type offers distinct advantages in generating humoral and cellular immune responses. However, each also has specific limitations. Inactivated and DNA vaccines may elicit weak immune responses against certain pathogens. mRNA vaccines demonstrate rapid and robust immunogenicity, but transient reactogenicity is commonly observed. Subunit vaccines provide enhanced specificity in humoral immunogenicity but often induce weak cellular immune responses. Viral vector vaccines can generate strong cellular immunity, yet their efficacy may be compromised by preexisting immunity to the vector. VLP vaccines effectively induce potent neutralizing antibodies. However, their complex manufacturing processes significantly impede large-scale production and clinical application.

**Figure 3 vaccines-13-01168-f003:**
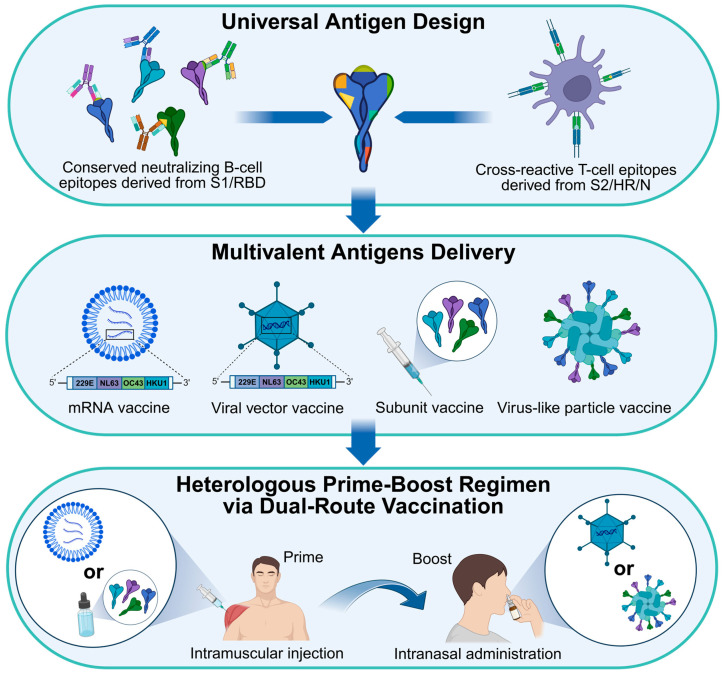
Strategic framework for the design and implementation of broad-spectrum vaccines targeting seasonal HCoVs. The structural- and computational-guided reverse vaccinology enables the rational design of universal antigens incorporating both conserved neutralizing B-cell epitopes derived from S1 and RBD antigens, and cross-reactive T-cell epitopes derived from S2, HR, and N antigens. These antigens are then multivalently displayed using optimized vaccine platforms. Finally, a heterologous prime-boost regimen administered via dual-route vaccination can be implemented to elicit robust systemic and mucosal immune responses.

**Table 1 vaccines-13-01168-t001:** Pathogenesis and epidemiology of seasonal HCoVs and available research models.

Coronavirus	HCoV-229E	HCoV-OC43	HCoV-NL63	HCoV-HKU1
Discovery (Year/Country)	1966/United States	1967/United States	2004/The Netherlands	2005/Hong Kong, China
Classification	Alphacoronavirus	Betacoronavirus	Alphacoronavirus	Betacoronavirus
Symptom	Coryza, rhinorrhea, nasal congestion, sore throat, cough, sneezing, malaise, headache, low-grade fever, myalgia	Sore throat, rhinorrhea, nasal congestion, cough, sneezing, malaise, headache, low-grade fever, myalgia	Rhinorrhea, fever, dry cough, wheezing, respiratory distress, coryza, croup	Rhinorrhea, fever, sore throat, cough, sputum production, dyspnea, chills, rigors, myalgia, headaches, febrile seizures
Incubation time (Day)	2–5	2–5	2–4	2–4
Epidemiology	Globally in winter	Globally in winter	Globally in winter	Globally in winter
Natural host	Bats	Rodents	Bats	Rodents
Intermediate host	Camels	Bovines	Unknown	Unknown
Evolutionary rate of Spike (substitutions/site/year)	7.34 × 10^−4^	8 × 10^−4^	3-6 × 10^−4^	4.39 × 10^−4^ (HKU1A) 1.46 × 10^−4^ (HKU1B)
Receptor	hAPN	9-*O*-Ac-Sia	hACE2	9-*O*-Ac-Sia, TMPRSS2
Tissue culture (Cell/Organoid)	WI-38, L-132, MRC-5, Huh7, Mv1Lu	BS-C-1, FT, RD, Mv1Lu, MRC-5, Huh7.5, HCT-8, HRT-18, IGROV-1	hACE2-overexpressing HEK293T, LLC-MK2, Vero-B4, Caco-2	HAE, 2D airway organoid
Mouse model	Ad5-hAPN-transduced *Ifnar1*^−/−^ mice	BALB/c mice, C57BL/6 mice	K18-hACE2 mice, Ad5-hACE2-transduced *Ifnar1*^−/−^ mice	Not available
Reference	[1,46,51,52,53,54,55,56,57,58,59]	[2,46,53,58,60,61,62,63,64,65,66,67,68,69,70,71,72,73,74,75]	[3,53,59,76,77,78,79,80,81,82,83,84,85,86]	[4,8,53,67,87,88,89,90,91,92,93]

**Table 2 vaccines-13-01168-t002:** Cross-reactivity of distinct coronavirus vaccines to seasonal HCoVs.

Vaccine Type	Vaccine Name	Target Virus	Reactivity to HCoV				Study Host	Reference
			229E	NL63	OC43	HKU1		
Inactivated Vaccine	γ-irradiated SARS-CoV-2	SARS-CoV-2	ND ^1^	ND	Yes *	ND	Mouse	[258]
	BBIBP-CorV	SARS-CoV-2	Yes	Yes	Yes	Yes	Human	[259]
	SinoVac	SARS-CoV-2	ND	Yes	No	ND	Human	[260]
DNA Vaccine	S/S.dCT/S.dTM/S1/RBD/S.dTM.PP ^2^	SARS-CoV-2	Yes	Yes	Yes	Yes	Rhesus Macaque	[262,263]
	SARS-CoV-2 S, M, or N	SARS-CoV-2	ND	ND	Yes	ND	Mouse	[168]
mRNA Vaccine	mRNA-1287 ^3^	HCoV-229E HCoV-NL63 HCoV-OC43 HCoV-HKU1	ND	ND	ND	ND	Human	[264]
	mRNA-OC43	HCoV-OC43	ND	ND	Yes *	ND	Mouse	[265]
	mRNA-1273	SARS-CoV-2	No	Yes *	Yes	Yes	Human, Mouse	[248,258,266,267]
	BNT162b2	SARS-CoV-2	Yes	Yes	Yes	Yes	Human, Mouse	[260,267,268,269,270]
Subunit Vaccine	HCoV-229E S	HCoV-229E	Yes	Yes	No	No	Mouse	[271]
	HCoV-NL63 S	HCoV-NL63	Yes	Yes	No	No	Mouse	[271]
	HCoV-OC43 S	HCoV-OC43	No	No	Yes	Yes	Mouse	[271]
	HCoV-HKU1 S	HCoV-HKU1	No	No	Yes	Yes	Mouse	[271]
	SARS-CoV-2 S	SARS-CoV-2	No	No	Yes	Yes	Mouse	[271]
	SARS-CoV-2 RBD or S	SARS-CoV-2	ND	ND	Yes	ND	Mouse	[258]
	NAIS-ag	HCoV-OC43	ND	ND	Yes *	ND	Rabbit	[272]
	CTR	SARS-CoV-2	ND	ND	ND	Yes	Mouse, Pigtail Macaque	[273]
VLP Vaccine	β-CoV mosaic_I53_dn5	β-CoVs	Yes	No	Yes	Yes	Mouse	[274]
	Ex_mosaic-NP	β-CoVs	ND	ND	Yes	Yes	Mouse	[275]
	VRP-229E-S	HCoV-229E	Yes *	ND	ND	ND	Mouse	[59,276]
	VRP-NL63-S	HCoV-NL63	ND	Yes *	ND	ND	Mouse	[59,276]
	RBD/HR_Ferritin	SARS-CoV-2	Yes *	ND	Yes *	ND	Mouse	[160]
	VLP-CoV S2	SARS-CoV	ND	No	Yes	Yes	Mouse	[277]
	VLP-CoV-2 S2	SARS-CoV-2	ND	Yes	Yes	Yes	Mouse	[277]
	S-I53-50 NP	SARS-CoV-2	Yes	Yes	Yes	Yes	Cynomolgus Macaque	[269]
Vector Vaccine	ChAdOx1-S-nCoV-19	SARS-CoV-2	Yes *	No	ND	No	Human	[266,278]
	FAdV-9-S19	SARS-CoV-2	Yes *	Yes *	Yes *	ND	Mouse, Cynomolgus Macaque	[279,280]
	Ad26.COV2.S	SARS-CoV-2	Yes	Yes	Yes	Yes	Rhesus Macaque	[263]
	Ad5-SARS-CoV-2 S	SARS-CoV-2	ND	ND	Yes	ND	Mouse	[258]
	Ad5-SARS-CoV-2 N	SARS-CoV-2	ND	ND	Yes	ND	Mouse	[258]
	VSV-SARS-CoV-2 S	SARS-CoV-2	ND	ND	Yes	ND	Mouse	[258]
	MVA-SARS-CoV S	SARS-CoV	ND	ND	Yes	ND	Mouse	[258]
	ChAd-BNT ^4^	SARS-CoV-2	Yes *	Yes *	Yes *	ND	Human	[281,282]

^1^ “ND” indicates “not detected”. ^2^ Rhesus macaques were immunized with DNA vaccines encoding S (*n* = 4), S.dCT (*n* = 4), S.dTM (*n* = 4), S1 (*n* = 4), RBD (*n* = 4), and S.dTM.PP (*n* = 5). Sera from these vaccinated animals were analyzed as one group and compared with the sham control group. ^3^ No preclinical result has been reported, and the effectiveness remains speculative. ^4^ ChAd-BNT is a heterologous prime-boost vaccination with sequential adenoviral-vectored ChA-dOx1-S-nCoV-19 vaccine and BNT162b2 mRNA vaccine. * Neutralizing cross-reactivity of the specific vaccine against the target coronavirus was observed. The asterisk-unmarked cross-reactivity indicated cross-binding activity.

## Data Availability

All data are contained within the article.

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
