# Peer review of "Current Status and Challenges of Vaccine Development for Seasonal Human Coronaviruses"

_vaccines, 2025, doi:10.3390/vaccines13111168_

Round 1
Reviewer 1 Report
Comments and Suggestions for Authors
The file is attached.

Author Response
Reviewer #1:
This manuscript by Zhang et al. is a timely and comprehensive review that synthesizes a vast body of knowledge on seasonal human coronaviruses (HCoVs). The authors provide a detailed overview of virology, pathogenesis, and the current landscape of vaccine development, with a valuable emphasis on cross-reactivity with SARS-CoV-2. The review is well-structured, the figures and tables are informative, and the strategic framework for future vaccines is a significant strength. However, the manuscript's focus occasionally drifts into general vaccinology and SARS- CoV-2-specific topics, diluting its core promise regarding seasonal HCoV-specific vaccine candidates. With revisions to sharpen this focus, add quantitative data on cross-reactivity and disease burden, and provide more concrete, actionable future directions, this manuscript has the potential to become an authoritative reference in the field. Overall, this is a high-quality and informative review that requires targeted revision to sharpen its focus and increase practical utility.
Reply: We gratefully acknowledge the reviewer for supporting our manuscript and sincerely appreciate the insightful suggestions provided to improve its quality. We have carefully revised the manuscript in response to the reviewer's recommendations, all of which have substantially enhanced the clarity, coherence, and completeness of our work.
Major comments
- Sharpening the focus
The review would benefit from a tighter focus on seasonal HCoVs. While the comprehensive coverage of vaccine platforms is valuable, approximately 40% of content of Sections 4 and 5 discusses non-seasonal HCoV examples, which dilutes the manuscript's core contribution.
Specific recommendations:
- Lines 72-103 (Introduction-Vaccine History): Condense from 32 lines to approximately 10 lines. The detailed history of BCG, cholera vaccines, polio, and HBV development, while interesting, is tangential. Retain only the conceptual evolution (live-attenuated → inactivated → subunit → structure-based design) with 1-2 brief examples per era. Move detailed examples to supplementary material if desired.
Reply: We gratefully acknowledge the reviewer's thoughtful suggestion. We agree that the original discussion of vaccine history was overly detailed. As a result, we have streamlined this section by removing extensive historical accounts and instead focusing on the conceptual evolution of major vaccine types, including live-attenuated, inactivated, subunit, and structure-based vaccines. For each category, we provide a concise example to illustrate its underlying principle and development.
- Lines 495-757 (Section 4-Vaccine Types): Currently 263 lines on general vaccine platforms. Consider: Reducing each platform subsection (4.1-4.6) by 30-40% by removing detailed non-coronavirus examples (e.g., lines 509-536 on cholera, typhoid, HAV, and rabies vaccines; lines 652-673 on diphtheria, tetanus, and HPV), replacing this space with expanded discussion of seasonal HCoV-specific applications for each platform, perhaps creating a supplementary table summarizing platform characteristics rather than extensive prose.
Reply: We thank the reviewer for this thoughtful suggestion. In the revised manuscript, we have substantially shortened the section on vaccine types. Specifically, we have removed vaccines targeting non-respiratory viruses, such as cholera, typhoid, HAV, rabies, diphtheria, tetanus, and HPV, and eliminated redundant descriptions of the development history for each vaccine. The reviewer also recommended that we retain or expand the discussion on seasonal HCoV-specific applications for each platform. However, currently available vaccines for seasonal human coronaviruses (HCoVs) remain extremely limited, which is precisely why we aim to highlight this gap through this review. We have thoroughly discussed the strategic framework for seasonal HCoV vaccine development in the Discussion section. Nevertheless, vaccines developed for other respiratory viruses, including influenza, SARS-CoV-2, MERS-CoV, and RSV, offer valuable precedents for advancing seasonal HCoV vaccine development. Accordingly, we have retained and expanded upon these successful examples in this section. We hope the reviewer finds this revised structure logically coherent and scientifically justified.
- Sections 5.1-5.4 (Vaccine Development): While cross-reactivity data is valuable, the balance is skewed:
~116 lines (799-915) on natural infection and inactivated vaccines (mostly SARS-CoV-2)
~85 lines (923-1007) on DNA/mRNA vaccines (mostly SARS-CoV-2 with limited seasonal HCoV data)
~81 lines (1008-1088) on subunit/VLP vaccines
~59 lines (1089-1148) on viral vector vaccines
Recommendation: Create clearer subsection divisions:
"Direct Seasonal HCoV Vaccines" (e.g., mRNA-1287, VRP-229E-S, VRP-NL63-S, mRNA-OC43, NAIS-ag)
"Cross-Reactive SARS-CoV-2 Vaccines" (can be condensed with brief overview; link to Supplement)
For each actual seasonal HCoV vaccine prototype, would be beneficial to provide if known: 1) Target virus and specific antigen (full S, RBD, S2, stabilization mutations), 2) Animal model used and key limitations, 3) Immunogenicity results with quantitative data (nAb titers, fold-increases, durability), 4) Homologous vs. heterologous protection data, and 5) current development stage.
Reply: We thank the reviewer for these insightful suggestions. We agree that the balance in the section titled "5. The Development of Vaccines Against Seasonal HCoVs" is skewed, and we apologize for not initially clarifying the rationale behind its structure. The primary aim of this section is to summarize all reported vaccine candidates specifically targeting seasonal HCoVs. However, following a systematic literature review, we found only a limited number of such candidates have been developed. In contrast, numerous studies on SARS-CoV-2 vaccines have reported cross-reactive immune responses against seasonal HCoVs. As a result, the discussion inevitably includes substantial information on this cross-reactivity, not because it was the original focus, but due to the scarcity of data on seasonal HCoV-specific vaccines. Recognizing the importance of this issue, we have now explicitly addressed this imbalance in the first paragraph of Section 5. The reviewer suggested subdividing each subsection into "Direct Seasonal HCoV Vaccines" and "Cross-Reactive SARS-CoV-2 Vaccines". However, for inactivated, DNA, and vector-based vaccines, only SARS-CoV or SARS-CoV-2 candidates have been reported to exhibit cross-reactivity with seasonal HCoVs, and no specific seasonal HCoV vaccines have been documented. For mRNA, subunit, and VLP vaccines, only 6 papers reported on seasonal HCoV vaccines. If the subsections were divided according to the proposed categories, a significant amount of data would be excluded due to insufficient studies. Therefore, given the limited number of reports on seasonal HCoV-specific vaccine candidates, we kindly request the reviewer's permission to present these findings collectively. For each existing seasonal HCoV vaccine prototype, the reviewer recommended providing detailed information on the target virus, specific antigen, animal model, quantitative outcomes, and protective efficacy. We fully agree that such data would clearly highlight both the availability of current evidence and the critical knowledge gaps. Accordingly, we have systematically re-evaluated all relevant vaccine studies and incorporated key details (if available), including the specific coronaviruses targeted; the primary antigens employed (e.g., RBD, S, S1, S2, M, or N); the animal models used in experiments (e.g., mouse, rabbit, macaque, or human); quantitative antibody responses, including titers and fold changes post-vaccination; and the nature of cross-reactive antibodies (i.e., cross-binding versus cross-neutralizing activity). These updated data also have been summarized and included in Table 2.
Introduction (Lines 46-118): Add 3-4 sentences explicitly stating: How this review differs from recent pan-coronavirus vaccine reviews (e.g., Moore et al. 2023, Vaccine; others focused on SARS-CoV- 2/MERS/SARS)? Why do seasonal HCoVs deserve dedicated attention despite mild disease (persistent global circulation, vulnerable populations, pandemic preparedness rationale)? What unique challenges do seasonal HCoVs present (distinct receptors, lack of models, limited market incliuentive)?
Reply: We thank the reviewer for this valuable suggestion. In the final paragraph of the Introduction section, we have revised the text to clearly articulate three key aspects: the rationale for developing broad-spectrum seasonal HCoV vaccines, how our review differs from existing ones, and the overall organization of our manuscript. First, we emphasize the importance of seasonal HCoV vaccine development by highlighting their potential to cause severe disease in vulnerable populations, their persistent global circulation, and their relevance to pandemic preparedness. Second, we clarify the distinguishing features of our review: while previous reviews primarily focus on highly pathogenic coronaviruses such as SARS-CoV and MERS-CoV, ours specifically addresses the challenges and current progress in vaccine development for seasonal HCoVs. We also reference the recent review by Moore et al. (Vaccine, 2023), which focuses on policy and funding strategies for pan-coronavirus vaccine initiatives, thereby further contextualizing our work. Finally, we provide a clear outline of the structure of our review and describe the literature search strategy employed. The reviewer also suggests including the unique challenges posed by seasonal HCoVs, such as distinct receptors, lack of suitable models, and limited market incentives. We have not expanded on these challenges in detail here, but provide comprehensive descriptions of them in the following sections "2. Pathogenesis and Epidemiology of Seasonal HCoVs" and in the Discussion.
Expected outcome: By implementing these changes, the manuscript should: Reduce overall length by ~15-20% (approximately 300-400 lines), increase seasonal HCoV-specific content from ~25% to ~50% of sections 4-5, provide a clearer evidence base for seasonal HCoV vaccine feasibility, and distinguish itself more clearly from existing pan-coronavirus reviews.
Rationale: Readers seeking information on seasonal HCoV vaccines will find current structure frustrating because actual seasonal HCoV vaccine candidates (HCoV-229E, -NL63, -OC43, -HKU1 specific) receive minimal dedicated discussion despite being the manuscript's stated focus. The strategic framework (Figure 3) is excellent but needs more concrete examples drawn from the limited existing seasonal HCoV vaccine literature to be actionable.
Reply: We sincerely thank the reviewer for the valuable suggestions, which have significantly enhanced the clarity and readability of our manuscript. In response, we have revised the manuscript accordingly by shortening the length of several sections and clarifying the descriptions of seasonal HCoV-specific and cross-reactive vaccines. The strategic framework presented in Figure 3 has been updated to include sources of conserved B-cell and T-cell epitopes, with further elaboration provided in the corresponding Discussion section. We hope that these revisions meet the reviewer's expectations.
- Expanding seasonal HCoV-specific vaccine data
The platform descriptions are accurate but generic. For each seasonal HCoV, a more systematic summary of all reported vaccine prototypes is needed. This should include the specific antigen(s) used, platform, animal model, and key immunogenicity results (differentiating binding from neutralizing antibodies and T-cell responses). If data is sparse for a particular virus, this should be explicitly stated as a critical research gap.
Specific recommendation: Consider adding a comprehensive summary table (similar to Table 2, but focused exclusively on seasonal HCoV-specific vaccines rather than cross-reactive ones) with columns for: target virus (229E/NL63/OC43/HKU1), vaccine name/designation, platform type, specific antigen(s) and modifications, animal model, immunogenicity data (binding Ab titers, nAb titers, T-cell responses), protection data (if available), current development stage, reference. This would make it immediately clear where data exists and where critical gaps remain.
Reply: We thank the reviewer for the thoughtful suggestion. In accordance with the reviewer's previous recommendations, we have conducted a more systematic summary of all reported vaccine prototypes, including information on target viruses, target antigens, vaccine types, animal models, key immunogenicity outcomes (such as cross-binding or cross-neutralizing antibodies), and antibody titers along with fold changes. Where data were not reported in the original studies, they were omitted from the summary. These summarized data have been incorporated into the revised Table 2. The reviewer also suggested including a new table focusing exclusively on seasonal HCoV-specific vaccines rather than cross-reactive ones. As mentioned above, only six studies have reported on seasonal HCoV vaccines. Given the limited number of available studies, a substantial amount of data would be excluded if cross-reactive vaccines were omitted. Therefore, we respectfully request to present these data in combination with cross-reactive findings to ensure a more comprehensive and informative synthesis.
- Quantifying cross-reactivity and reconciling conflicts
The section on cross-reactivity is a highlight but would be strengthened if authors would consider the inclusion of quantitative data, such as sequence identity of conserved epitopes (S2, N) and fold-changes in neutralization titers.
Specific recommendations:
- The review would benefit if authors would add quantitative epitope data: For conserved regions mentioned (S2 subunit, N protein, specific epitopes like residues 811-830, 1014-1030, 1144-1163), provide: Percent sequence identity across seasonal HCoVs and SARS-CoV-2, a small supplementary table or figure panel showing sequence alignments of key epitopes, HLA restriction data for T-cell epitopes where available.
- Quantify cross-reactivity magnitude if available: When discussing cross-reactive antibodies (e.g., lines 807-826, 965-1007), include: fold-change in titers (e.g., "increased 2.5-fold" rather than just "increased"), baseline vs. post-infection/vaccination absolute titers where available, neutralization IC50 or EC50 values for cross-reactive responses.
Reply: We thank the reviewer for these thoughtful suggestions. The reviewer recommended including quantitative epitope data and cross-reactivity magnitude data for each antigen and vaccine, respectively. Specific epitopes, such as those spanning residues 811-830, 1014-1030, and 1144-1163, have been reported as conserved regions that elicit cross-reactive B-cell and T-cell responses, and these are already mentioned in our review. However, due to limitations in data availability and the scope of our review, we did not perform a quantitative analysis of epitope data, including sequence alignments of key epitopes or HLA restriction information for T-cell epitopes. While these epitopes are inferred to contribute to cross-reactivity, our review was not designed to conduct secondary data analyses based on primary studies, but rather to synthesize and summarize existing findings. Regarding the suggestion to include cross-reactivity magnitude data, we have now incorporated such data for each vaccine type. Both antibody titers and fold changes are described for each vaccine, wherever available. Specifically, we have distinguished between cross-binding and cross-neutralizing antibodies in Table 2 and added information on the experimental contexts in which cross-reactivity was measured. We hope the reviewer will find this presentation of quantitative cross-reactivity data acceptable in lieu of comprehensive quantitative epitope analyses.
Reconcile contradictions explicitly: The manuscript presents conflicting data on whether preexisting immunity to seasonal HCoVs is beneficial or detrimental to SARS-CoV-2 outcomes (lines 845-856 vs. 267-270, 831-844). While some studies show reduced COVID-19 severity with prior seasonal HCoV exposure, others demonstrate increased susceptibility or impaired vaccine responses. This contradiction requires explicit reconciliation with a mechanistic explanation.
Suggested additions:
3.1 Antigenic sin (immune imprinting) as a unifying mechanism: The authors should discuss how original antigenic sin may explain these contradictory findings. Upon SARS-CoV-2 exposure or vaccination, preexisting memory B cells targeting conserved S2/N epitopes may be preferentially recalled, leading to: 1) Rapid production of cross- reactive but non-neutralizing antibodies, 2) Competitive suppression of naïve B cell responses against protective RBD epitopes, 3) The "back-boosting" phenomenon described (lines 262-275, 804-826) without generating protective immunity against the new pathogen
3.2 Quality versus quantity of cross-reactive responses: The conflicting outcomes likely reflect differences in: Beneficial scenarios: Low-level preexisting cellular immunity (CD4+/CD8+ T cells targeting conserved epitopes) providing partial protection through viral clearance and enhanced germinal center responses (lines 858-880, 905-915, 1287- 1294) or Detrimental scenarios: High-titer non-neutralizing antibodies that:
- Saturate Fc receptors without effective viral neutralization
- Redirect immune responses away from protective RBD epitopes (lines 855-858, 1226-1230)
- Potentially form immune complexes contributing to immunopathology
- Impair de novo generation of SARS-CoV-2-specific neutralizing antibodies (line 851-856)
3.3 Study design heterogeneity: The authors should acknowledge that contradictions may also arise from: 1) Retrospective vs. prospective designs: Different ability to control for confounders, 2) Cohort characteristics: Age distribution, comorbidities, timing and frequency of seasonal HCoV exposures 3) Timing of assessment: Early infection (where T-cell immunity may be protective) vs. chronic/severe disease (where non- neutralizing antibodies may be detrimental) 3) Epitope specificity measured: Studies measuring S1/RBD responses vs. S2/N responses reach different conclusions
3.4 Critical implications for vaccine design (directly relevant to lines 1265-1294):
- How should conserved epitopes be incorporated to leverage beneficial T-cell immunity while avoiding detrimental B-cell imprinting?
- Would sequential vaccination strategies (e.g., conserved epitopes first to establish T-cell help, followed by variable RBD epitopes) overcome immune focusing?
- Should the proposed "universal antigen" design (lines 1265-1294) deliberately minimize exposure of immunodominant non-neutralizing conserved epitopes while preserving T-cell epitopes?
- Could structure-guided masking of conserved non-neutralizing B-cell epitopes combined with preserved T-cell epitopes address this challenge?
3.5 Recommended citations: The following papers provide relevant frameworks:
|
Reference |
Key Concept |
Main Finding Relevant to Imprinting |
|
Turner et al., Nature 2021 SARS- CoV-2 infection induces long-lived bone marrow plasma cells in humans |
Long-lived Plasma Cells |
mRNA vaccination establishes a persistent population of cells producing antibodies against the original strain, creating a stable "baseline" response. |
|
Röltgen et al. Immune imprinting, breadth of variant recognition, and germinal center response in human SARS- CoV-2 infection and vaccination |
B Cell Recall Bias |
The immune response to a new variant (Omicron) is dominated by the recall of cross-reactive memory B cells from prior exposure, limiting the response to new epitopes. |
|
Laidlaw & Ellebedy The germinal centre B cell response to SARS- CoV-2 |
Germinal Center Dynamics |
Provides the overarching framework: the germinal center is the key site where imprinting occurs, as pre-existing memory B cells outcompete naive ones for resources. |
Conclusion: The authors' proposed strategies (lines 1265-1332) would be substantially strengthened by explicitly addressing how antigenic sin will be avoided or mitigated. A clear statement on whether conserved epitopes should be avoided, carefully balanced, or strategically presented in the context of immune imprinting is essential for guiding future pan- coronavirus vaccine development.
Reply: We sincerely thank the reviewer for the thoughtful suggestions and valuable guidance regarding the development of universal antigen design. The reviewer's comments are grounded in the observation that contradictory findings have been reported regarding the impact of preexisting seasonal HCoV immunity. Some studies associate it with enhanced disease severity, while others report reduced severity. These discrepancies warrant careful and comprehensive explanation. Furthermore, the design of universal HCoV antigens must take into account the potential influence of heterogeneous preexisting immunity to specific HCoVs. In response, we have addressed these issues in the Discussion section.
Firstly, we have incorporated the reviewer's insightful recommendations into the subsection of the Discussion titled "6.1. Current Challenges in Seasonal HCoV Vaccines". We have identified preexisting immunity to seasonal human coronaviruses (HCoVs) as the sixth major challenge in developing seasonal HCoV vaccines. Existing studies present conflicting findings: preexisting immune responses to seasonal HCoVs have been associated with either increased disease severity in COVID-19 patients or altered susceptibility to SARS-CoV-2 infection in animal models. The reviewer proposed three potential explanations for this inconsistency. In response, we have revised this section by incorporating the following clarifying text: The antigenic sin, also known as immune imprinting, could be a unifying mechanism underlying the increased disease severity of COVID-19 in patients with preexisting immune responses to seasonal HCoVs (Relevant references provided by the reviewer: PMID: 34030176; PMID: 35148837; PMID: 34873279). Upon SARS-CoV-2 exposure or vaccination, preexisting memory B cells targeting conserved S2 or N epitopes of HCoVs may be preferentially recalled, leading to the rapid generation of cross-reactive but non-neutralizing antibodies, thereby competitively suppressing naïve B-cell responses against protective RBD epitopes, consequently resulting in the back-boost phenomenon without generating protective immunity against specific HCoVs. These conflicting reports also demonstrate the importance of balancing the quality and quantity of cross-reactive immune responses against seasonal HCoVs. The conflicting outcomes likely reflect differences in both beneficial and detrimental scenarios. The low-level preexisting CD4+ and CD8+ T-cell immunity targeting conserved epitopes may provide partial protection through viral clearance and enhanced germinal center responses. However, high-titer non-neutralizing antibodies may also saturate Fc receptors without effective viral neutralization, redirect immune responses away from protective RBD epitopes, impair de novo generation of SARS-CoV-2-specific neutralizing antibodies, and potentially form immune complexes contributing to immunopathology. Of note, the study design heterogeneity may also contribute to these contradictions. Retrospective versus prospective study designs differ in their ability to control for confounding factors. Key cohort characteristics, such as age distribution, comorbidities, and the timing and frequency of seasonal HCoV exposures, can influence study outcomes. The timing of immune assessment is also critical. Early infection stages, where T-cell immunity may be protective, contrast with chronic or severe disease phases, where non-neutralizing antibodies might contribute to immunopathology. Furthermore, differences in epitope specificity across studies, particularly those measuring S1 or RBD responses compared to those targeting S2 or N proteins, may explain divergent conclusions.
Secondly, we have discussed the critical implications of the aforementioned influences on vaccine design, which are addressed in the first paragraph of the subsection entitled "6.2. Strategic Framework for the Development of Seasonal HCoV Vaccines". The reviewer raised four concerns regarding whether these antigen design strategies could enhance vaccine breadth and circumvent immune imprinting. We have incorporated discussions of these points into our revised manuscript. However, rather than providing detailed elaborations, we have outlined these potential approaches. A comprehensive explanation of these methods is important, as is acknowledging the current limitations in antigen design. We propose these strategies to guide future efforts toward developing universal antigens. Given the current limitations in knowledge and technical accessibility regarding these methodologies, we kindly request that the reviewer permit us to present these proposals in a conceptual manner. Our revised description has been outlined as follows: The S proteins of seasonal HCoVs contain both neutralizing B‑cell epitopes and HLA‑promiscuous T‑cell epitopes, making them a predominant target for universal vaccine antigen design. By integrating structural insights from antigen-antibody complexes with computational consensus sequence analyses, a rationally designed de novo antigen should incorporate conserved neutralizing B-cell epitopes and cross-reactive T-cell epitopes, while minimizing the exposure of strain‑specific, immunodominant decoy epitopes that elicit narrow or non-protective immune responses. Typical targets include conserved conformational epitopes within RBD of the S1 subunit and cross-reactive HR regions in the S2 subunit. Conserved T-cell peptides derived from the N and ORF proteins can also be incorporated into antigen designs. Importantly, vaccine antigens must minimize exposure or immunodominance of conserved non-neutralizing epitopes that may misdirect B cell responses toward ineffective antibody production. Thus, rational antigen scaffold design and structural stabilization are essential strategies to focus immune responses on protective epitopes and finalize optimized universal vaccine antigens. Additionally, a sequential vaccination strategy, initiating with conserved epitopes to establish T-cell help, followed by administration of variable RBD epitopes, could serve as an alternative approach to mitigate immune focusing.
- Clarifying animal model limitations
The limitations of existing animal models, especially for HCoV-HKU1 and HCoV-NL63, represent a critical barrier to vaccine development that deserves more prominent discussion. The current treatment (primarily in Table 1 and brief mentions in Section 2) understates the severity of this problem. Why this matters: 1) Regulatory path: Without validated animal models demonstrating protection against authentic virus challenge, vaccine candidates face significant regulatory hurdles. The FDA and EMA typically require challenge studies in relevant animal models before advancing to clinical trials for respiratory pathogens. 2) Vaccine evaluation: The inability to assess protective efficacy (viral load reduction, disease prevention) means that immunogenicity data alone must guide development decisions, increasing the risk of clinical failure. Correlates of protection cannot be established without challenge models. 3) Development timeline: The absence of standardized models means each research group must invest 1-2 years developing and validating their own systems before vaccine testing can begin, fragmenting the field and delaying progress.
Reply: We sincerely thank the reviewer for the insightful summary regarding the limitations of current animal models. We fully agree that, in the absence of appropriate animal models capable of demonstrating protection against authentic viral challenge, vaccine candidates encounter significant regulatory challenges. Furthermore, evaluating specific seasonal HCoVs, particularly HCoV-HKU1, would remain unfeasible without such models. Establishing correlations of protection is inherently dependent on challenge studies, which are currently lacking. Moreover, the absence of validated animal models not only prolongs the vaccine development timeline but also fragments research efforts, thereby impeding overall progress. Given the importance of this observation, we have incorporated these points into the Discussion section.
Specific recommendations:
For HCoV-HKU1 (Lines 283-289, Table 1): The text states "no available animal models" while referencing the chimeric MHV system (line 284-289). This is contradictory and confusing. Clarify explicitly: 1) No wild-type mouse or transgenic mouse model exists for authentic HCoV-HKU1, HAE and 2D organoids support replication but are impractical for large-scale vaccine studies (specify why: cost, throughput, lack of systemic immunity assessment), 2) The chimeric J2.2-MHV system (ref 116) is a surrogate model that allows study of individual HKU1 proteins but does not recapitulate natural infection, 3) This represents a critical gap requiring urgent research investment (e.g., development of mouse-adapted strains via serial passage, as done for OC43 and SARS-CoV-2).
Reply: We thank the reviewer for pointing out this contradictory and providing the correct recommendation. We have added the reviewer's kind suggestion in "2.4. HCoV-HKU1" section. The description is as follows: "No wild-type mouse or transgenic mouse model exists for authentic HCoV-HKU1. Human ciliated airway epithelial cell cultures (HAE) and two-dimensional (2D) airway organoids have been successfully used to support HCoV-HKU1 replication. However, their complex and resource-intensive production procedures have substantially limited their widespread application in vaccine evaluation." "However, the chimeric MHV (J2.2) system is a surrogate model that allows study of individual HKU1 proteins but does not recapitulate natural infection. This represents a critical gap requiring urgent research investment, including the development of mouse-adapted strains via serial passage, which has been successfully applied to HCoV-OC43 and SARS-CoV-2."
For HCoV-NL63 (Lines 231-237, Table 1): The manuscript contains an internal inconsistency that must be resolved: 1) Table 1 lists "K18-hACE2 mice, Ad5-hACE2-transduced IFNAR−/− mice" as available models 2) Line 234-236 states: "Ad5-hACE2-transduced wild-type mice nor hACE2-knockin (KI) mice support HCoV-NL63 infection"
Clarify:
- K18-hACE2 transgenic mice ARE susceptible (ref 86, Bentley et al. 2024)
- Ad5-hACE2 transduction works ONLY in IFNAR−/− mice, NOT in wild-type mice (ref 58, Liu et al. 2023)
- The requirement for IFNAR−/− background limits translatability to human immunity
- Specify whether pseudotyped virus neutralization assays can partially substitute for authentic virus studies (currently unclear)
Reply: We sincerely apologize for the inconsistent description. K18-hACE2 transgenic mice are indeed susceptible to infection, as demonstrated by Bentley et al. (2024) and Liu et al. (2023). Notably, Liu et al. (2023) employed two distinct hACE2 mouse models, referred to as "K18-hACE2 mice" and "hACE2-KI mice", and reported that hACE2-KI mice did not support HCoV-NL63 infection. To avoid confusion and clearly emphasize that certain hACE2 mouse models can be used for HCoV-NL63 infection studies, we have removed the description to hACE2-KI mice, which could otherwise mislead readers regarding the broader utility of hACE2-expressing mice.
Authors should consider adding a dedicated subsection (2-3 paragraphs) in the challenges section (around line 1262) titled: "Critical need for standardized animal models." This should: Explicitly state that HCoV-HKU1 remains the only seasonal HCoV without a practical animal model: 1) Discuss why this is a "chicken-and-egg" problem: vaccine development is deprioritized because models don't exist, but model development is deprioritized because disease is perceived as mild. 2) Propose possible solutions: international consortium for model development, prioritization of mouse-adapted strain generation, validation of organoid systems for specific endpoints (neutralization correlates vs. full protection studies).
Reply: We thank the reviewer for the thoughtful suggestion. In light of the reviewer's earlier recommendations, we have revised the description of the first challenge in developing seasonal HCoV vaccines, the lack of simplified tissue culture systems and animal models for HCoV-HKU1. We have also proposed potential solutions, including the development of a mouse-adapted strain and the use of organoid systems to assess specific endpoints.
In the strategic framework discussion (Lines 1332-1376): Add explicit acknowledgment that the proposed prime-boost strategies cannot currently be validated for HCoV-HKU1, and that model development must occur in parallel with antigen design.
Reply: We thank the reviewer for this kind suggestion. We have incorporated an explicit statement in the discussion section regarding the heterologous prime-boost regimen. The revised text reads as follows: "Notably, the proposed heterologous prime-boost strategies cannot currently be validated for HCoV-HKU1, necessitating concurrent development of mouse models alongside antigen design."
Expected outcome: These changes would transform a peripheral technical issue into a recognized central challenge, potentially motivating funding agencies and researchers to prioritize model development as a field-wide need.
Reply: We gratefully acknowledge the reviewer for the insightful suggestions provided. We have incorporated necessary descriptions and carefully revised each section to enhance clarity and completeness.
- Strengthening public health rationale
The justification for developing vaccines against typically mild viruses needs substantial bolstering with quantitative epidemiological and economic data. Currently, the manuscript relies primarily on qualitative statements (e.g., "substantial proportion," "significant pathogenicity") without providing the quantitative burden- of-disease data needed to justify the considerable R&D investment required.
Specific data gaps to address:
- Pandemic Preparedness Rationale (expand Lines 63-71):
The "future zoonotic spillover" argument needs strengthening: 1) Explicitly state that seasonal HCoV vaccines provide a "platform" for rapid response to novel coronaviruses 2) Quantify if possible the zoonotic threat: X bat coronaviruses with potential for spillover (line 67 mentions ">4,000 distinct coronaviral sequences") 3) Make the economic argument: Investing in seasonal HCoV vaccines now may cost $X million but could save $X billion if it accelerates response to the next SARS-CoV-3 4) Reference the "prototype pathogen" approach advocated by WHO and CEPI.
Reply: We thank the reviewer for this kind suggestion. We have clearly stated that seasonal HCoV vaccines could provide a "platform" for rapid response to novel coronaviruses, which could be a priority to develop pan-coronavirus vaccines against currently circulating strains as well as potentially future "X" coronaviruses. The reviewer also suggested including how many bat coronaviruses with potential for spillover within the 4,000 distinct coronaviruses. However, when we searched the original paper, the authors did not elucidate this prediction. While another paper reported by Grange, Z.L. et al. (Proc. Natl. Acad. Sci. USA 2021) demonstrated that among the top 50 wildlife viruses with zoonotic spillover potential, 21 viruses belong to coronaviruses. This paper has been cited and summarized in the first paragraph of the Introduction section. The reviewer also recommended incorporating the economic argument and the prototype pathogen approach. However, due to insufficient global epidemiological data on seasonal HCoVs and the limited number of reported viral sequences, we are unable to fully develop the economic argument or adequately apply the "prototype pathogen" approach advocated by WHO and CEPI. Although these limitations have been acknowledged in the Discussion section, they should be carefully addressed in future research.
- Comparative risk-benefit analysis (possibly add new section, ~150-200 words):
Address the elephant in the room: Why vaccinate against typically mild infections? 1) Precedent: We vaccinate against chickenpox (varicella), which is typically mild in children but severe in adults and immunocompromised - provide comparative CFRs 2) Vulnerable populations: Even if 95% of infections are mild, the remaining 5% in high-risk groups may represent millions of severe cases globally 3) Herd immunity potential: Could vaccination reduce transmission to vulnerable populations who cannot be vaccinated? 4) Long-term sequelae: Are there any data on post-viral syndromes after seasonal HCoV infection? (If not, state this as a research gap).
Suggested structure: Consider adding a new Section 2.6: "Public health impact and rationale for vaccine development" that consolidates this quantitative data. This would appear after Section 2.5 (Comparison with Highly Pathogenic HCoVs) and before Section 3 (Structures and Antigens) Expected outcome: These additions would transform the manuscript from a primarily scientific/technical review into one that also makes a compelling public health and economic case for seasonal HCoV vaccine development. This is essential for:
- Motivating funding agencies to prioritize this area
- Convincing pharmaceutical companies that market opportunity exists
- Providing vaccine developers with target product profiles
- Helping policymakers understand why this deserves attention despite mild typical disease
If quantitative data is limited: The manuscript should explicitly state which data gaps exist and call for epidemiological studies to fill them. This is itself a valuable contribution.
Reply: We thank the reviewer for the thoughtful suggestions. In accordance with the reviewer's recommendation, we have added a new section titled "2.6. Public Health Impact and Rationale for the Development of Seasonal HCoV Vaccine" at the end of Section 2. We agree that addressing the public health burden of seasonal HCoVs and articulating the rationale for vaccine development are essential. However, due to limited availability of epidemiological data, quantitative evidence, and information on long-term sequelae, we have refrained from providing extensive elaboration. Instead, we explicitly acknowledge these knowledge gaps and emphasize their importance for future research. Furthermore, we highlight the potential impact of seasonal HCoVs on high-risk populations and discuss the prospect of achieving vaccination-mediated herd immunity. Addressing these gaps will be crucial for advancing vaccine development efforts, which requires coordinated support from funding agencies, pharmaceutical companies, vaccine developers, and policymakers. We have accordingly revised our manuscript as follows: Due to the relatively mild symptoms associated with seasonal HCoV infections and the limited availability of comprehensive epidemiological data on their global endemicity, vaccine development against seasonal HCoVs has historically been deprioritized. This review provides a comprehensive summary of the virological characteristics, epidemiology, and current vaccine development status of seasonal HCoVs. Nevertheless, the development of virus-specific or broad-spectrum vaccines remains crucial for public health preparedness. Varicella-zoster virus (VZV), which causes chickenpox, typically results in mild disease in children but can lead to severe complications in adults and immunocompromised individuals (PMID: 40872869). The successful development and implementation of VZV vaccines have significantly reduced both disease burden and associated morbidity (PMID: 40733707). Similarly, although 95% of seasonal HCoV infections are mild, the remaining 5% in high-risk groups may represent millions of severe cases globally. Thus, developing seasonal HCoV vaccines is urgently needed for vulnerable populations. Moreover, vaccinating the general population against seasonal HCoVs could reduce transmission to those who cannot be vaccinated, potentially conferring herd immunity. Another significant data gap regarding seasonal HCoV infections is the potential for long-term sequelae, a phenomenon observed in SARS-CoV-2 infections (PMID: 36357676). Whether seasonal HCoVs can lead to post-viral syndromes remains unknown. Post-infection follow-up studies are essential to address this question. Filling these knowledge gaps is critical for advancing efforts toward vaccine development, as it would help motivate funding agencies to prioritize research in this area, demonstrate to pharmaceutical companies the existence of a viable market opportunity, provide vaccine developers with clearly defined target product profiles, and assist policymakers in recognizing the importance of seasonal HCoV vaccines despite the typically mild nature of these infections.
- Enhancing figures and future directions
Figure 1: Please use a consistent genomic scale across all four viruses, annotate accessory proteins uniformly (some are labeled, others are not), and clearly indicate S1/S2 boundaries on the genomic maps. Consider adding a scale bar showing kb or nucleotide positions.
Reply: We thank the reviewer for the kind suggestions. In the revised manuscript, we have implemented a uniform genomic scale for all four viruses. We have also consistently annotated all genes, with particular attention to accessory proteins. Besides, the S1/S2 cleavage site has been clearly marked within each Spike gene. Additionally, we provide a schematic illustration of the virus depicting its various structural proteins, along with a cross-sectional view that reveals the viral genome and associated proteins.
Figure 3 (Strategic framework): This excellent conceptual figure would be even more powerful if it included examples of conserved or universal antigens at each stage (e.g., "S2 stem helix mosaic," "RBD + HR1 chimera," "N protein CTL epitopes").
Reply: We gratefully acknowledge the reviewer's thoughtful suggestion. In response, we have incorporated conserved or universal antigens from which the epitopes are derived, including conserved neutralizing B-cell epitopes sourced from S1 and RBD antigens and cross-reactive T-cell epitopes derived from S2, HR, and N antigens. These examples have been clearly described in the Discussion section to better highlight their relevance and implications, and also showed in Figure 3.
Future directions dection (Lines 1263-1391): Currently titled "Conclusions" but primarily discusses challenges. Consider restructuring as: Section 6.1: "Current Challenges" (existing content, lines 1263-1330) and Section 6.2: "Strategic Framework for Vaccine Development" (existing content, lines 1332-1376, expanded).
Reply: We agree with the reviewer that the current Discussion section lacks a clear structure. To improve clarity and logical flow, we have reorganized this section into two distinct subsections: "6.1. Current Challenges in Seasonal HCoV Vaccines" and "6.2. Strategic Framework for the Development of Seasonal HCoV Vaccines".
- Methodological transparency
Adding a brief "Literature search strategy" section (100-150 words, could be placed after Introduction or as first paragraph of Methods section if one is added) would enhance the review's reproducibility and rigor. This should include: 1) Databases searched (PubMed, Web of Science, Google Scholar, preprint servers), 2) Keywords used (e.g., "seasonal coronavirus," "HCoV-229E," "HCoV vaccine," "common cold coronavirus"), 3) Date range (e.g., 1966-2025, or specific cutoff date), 4) Inclusion/exclusion criteria, 5) Number of papers initially identified vs. finally included, 6) Whether systematic review guidelines (PRISMA) were followed.
This is increasingly expected for comprehensive reviews in high-quality journals.
Reply: We thank the reviewer for the kind suggestion. The literature search strategy has been incorporated into the final paragraph of the Introduction, thereby enhancing the reproducibility and methodological rigor of our review. Major strategies were described as follows: The literature search strategy encompassed databases from PubMed, Web of Science, Google Scholar, and preprint servers. Search keywords included "seasonal coronavirus", "common cold coronavirus", "HCoV vaccine", and "coronavirus vaccine". The search covered publications dated from 1960 to 2025.
Minor Comments
Abbreviations and terminology. Please define all abbreviations at first use throughout the manuscript. Examples currently missing or inconsistently used:
HAE (human airway epithelial cells) - first used line 279 without definition
HE (hemagglutinin-esterase) - defined in Figure 1 but not in text
HR1/HR2 (heptad repeat 1/2) - used line 438 without definition
9-O-Ac-Sia - first used line 183, defined line 272
ERGIC, RNP, PLpro, Mpro - verify all are defined at first text use
Reply: We thank the reviewer for pointing out this omission and we have defined all abbreviations at first use.
Typographical errors
Page 2, Line 51: "score or scratchy throat" should be "sore or scratchy throat"
Page 6, Line 260: "To data" should be "To date"
Line 58: Verify SARS outbreak dates. Most sources cite 2002-2003, but some extend to 2004 for final case clearance. Clarify whether referring to outbreak period or final case.
Reply: Thanks for pointing out these mistakes and we have corrected them. The final case clearance of SARS-CoV should indeed be extended to 2004, as has been corrected in our revised manuscript.
Consistency issues
Nomenclature: Check consistency of viral protein naming conventions (e.g., "Spike protein" vs. "S protein" vs. "spike"). Choose one primary term and use consistently.
Reply: We thank the reviewer for highlighting this inconsistency. We have now used "S protein" consistently. Upon first use, we defined "Spike protein" as "S protein".
Reference formatting: Ensure all references follow journal guidelines consistently. Spot-check revealed minor inconsistencies in journal name abbreviations and page number formatting.
Reply: We thank the reviewer for pointing out these inconsistencies. We have carefully checked all the references and corrected the journal name abbreviations and page number formatting.
Content additions
Recent literature: The reference list is comprehensive through early 2024. Consider adding key papers from late 2024 and early 2025 if available, particularly: Any new seasonal HCoV vaccine candidates, Updated epidemiological data, Recent correlates of protection studies, Advances in HCoV-HKU1 model development.
Reply: We thank the reviewer for this valuable suggestion. In our revised manuscript, we have incorporated more than 30 recent and relevant publications from 2024 to 2025. Specifically, we included three studies on seasonal HCoV vaccine candidates (PMID: 40462906, PMID: 40699754, preprint doi.org/10.1101/2025.07.16.665240). However, due to the limited availability of epidemiological data, recent correlations of protection studies, and advances in HCoV-HKU1 model development, we were unable to identify additional relevant publications to include at this time.
Table and figure improvements
Table 1: Consider adding a column for "Clinical attack rate" or "Reinfection frequency" to better characterize transmission dynamics and duration of immunity.
Reply: Thank the reviewer for the kind suggestion. However, data on clinical attack rates and reinfection frequencies for each individual seasonal HCoV remain limited. Most studies indicate that the four seasonal HCoVs collectively account for 15-30% of all common cold cases, as noted in the Introduction. Furthermore, reinfections with seasonal HCoVs are common, with documented cases as early as 6 months after primary infection and the majority occurring approximately 12 months post-infection. This report has been described in "6.1. Current Challenges in Seasonal HCoV Vaccines" section.
Table 2: Excellent compilation. Consider adding:
Symbol or color-coding to distinguish binding vs. neutralizing cross-reactivity (Symbols denote binding (○) vs neutralizing (●)).
Magnitude of effect where available (fold-change in titers ).
Visual hierarchy to highlight most promising vaccine candidates.
Reply: We thank the reviewer for these suggestions. In our revised Table 2, we have used (*) to indicate neutralizing cross-reactivity of the specific vaccine against the target coronavirus. The asterisk-unmarked cross-reactivity indicated cross-binding activity. However, we were unable to report the magnitude of the effect due to a lack of precise comparative data. Given that most vaccines exhibited cross-binding activity but elicited only weak cross-neutralizing antibody responses, it remains difficult to identify the most promising vaccine candidates. We do apologize for the omission of this important point.
With these revisions, this manuscript has excellent potential to serve as the definitive reference on seasonal HCoV vaccine development and to catalyze renewed interest in this neglected area. The strategic framework (Figure 3) is particularly valuable and, with the suggested enhancements, could guide the field for years to come.
The authors have clearly invested substantial effort in this comprehensive review. The recommended revisions, while extensive, are focused on enhancing the manuscript's impact and practical utility rather than addressing fundamental flaws. I look forward to reviewing the revised version.
Reply: We gratefully acknowledge the reviewer for their insightful comments and suggestions. We have carefully revised the entire manuscript to incorporate all recommended improvements. We hope that the revised version meets the reviewer's expectations and satisfies the journal's standards.
Reviewer 2 Report
Comments and Suggestions for Authors
The article's topic is pertinent, and its content may be beneficial to readers of the Vaccine. Four common human coronaviruses (HCoV), including two alpha types (HCoV-NL63 and HCoV-229E) and two beta types (HCoV-HKU1 and HCoV-OC43), commonly cause mild upper respiratory tract diseases. However, in vulnerable populations, they can lead to more severe diseases such as pneumonia. There are no standardized vaccines against these viruses, and only limited antiviral therapy options are available to treat the most severe cases. Overall, I find the results of this review article to be highly valuable. The development, testing, and implementation of HCoV vaccines in medical practice is a pressing concern for healthcare systems worldwide. Please find below some comments and suggestions for your consideration.
(1) In section 2, the authors presented a separate description of 4 HCoV. Concurrently, it would be advisable to present a generalizing subsection describing both the general and distinctive features of HCoV. For instance, HCoV-229E and HCoV-OC43 viruses have been shown to generate new variants and achieve widespread intercontinental distribution due to continuous genetic drift, recombination, and complex migration patterns (doi: 10.1038/s44298-024-00058-w). However, a particular concern is that modern HCOVs differ from laboratory-adapted reference strains that are used to study the biology of the virus and evaluate medical countermeasures (doi: 10.1128/jvi.00684-25). Furthermore, there is concern about the possibility of increased pathogenicity of various infections in co-infection with certain HCoV. There is also concern about the possibility of genetic recombination and the emergence of new variants and strains of these viruses with a higher level of pathogenicity and virulence.
(2) Section 5. The conclusions, which span six pages, align more closely with the discussion section. It would be advisable for the authors to provide a concise summary of their work, no more than one page in length, in the section designated as "Conclusions." (Section 6).
Author Response
Reviewer #2:
The article's topic is pertinent, and its content may be beneficial to readers of the Vaccine. Four common human coronaviruses (HCoV), including two alpha types (HCoV-NL63 and HCoV-229E) and two beta types (HCoV-HKU1 and HCoV-OC43), commonly cause mild upper respiratory tract diseases. However, in vulnerable populations, they can lead to more severe diseases such as pneumonia. There are no standardized vaccines against these viruses, and only limited antiviral therapy options are available to treat the most severe cases. Overall, I find the results of this review article to be highly valuable. The development, testing, and implementation of HCoV vaccines in medical practice is a pressing concern for healthcare systems worldwide. Please find below some comments and suggestions for your consideration.
Reply: We gratefully acknowledge the reviewer for their constructive feedback and support of our manuscript. We have carefully considered all comments and have revised the manuscript accordingly.
(1) In section 2, the authors presented a separate description of 4 HCoV. Concurrently, it would be advisable to present a generalizing subsection describing both the general and distinctive features of HCoV. For instance, HCoV-229E and HCoV-OC43 viruses have been shown to generate new variants and achieve widespread intercontinental distribution due to continuous genetic drift, recombination, and complex migration patterns (doi: 10.1038/s44298-024-00058-w). However, a particular concern is that modern HCOVs differ from laboratory-adapted reference strains that are used to study the biology of the virus and evaluate medical countermeasures (doi: 10.1128/jvi.00684-25). Furthermore, there is concern about the possibility of increased pathogenicity of various infections in co-infection with certain HCoV. There is also concern about the possibility of genetic recombination and the emergence of new variants and strains of these viruses with a higher level of pathogenicity and virulence.
Reply: We gratefully acknowledge the reviewer for this valuable suggestion. We fully agree that it is important to include a dedicated subsection outlining the "General and Distinctive Features of Seasonal HCoVs". In response to feedback from another reviewer, we have relocated the original Subsection 2.5, titled "Comparison Between Seasonal and Highly Pathogenic HCoVs", to the penultimate paragraph of the Introduction. Concurrently, we have replaced the original Subsection 2.5 with a newly structured section entitled "2.5. General and Distinctive Features of Seasonal HCoVs", which now incorporates comprehensive content expanded in accordance with the reviewer's comments. In this subsection, we first summarize the general genetic features of the four seasonal HCoVs. We then examine the potential for antigenic drift and recombination in HCoV-229E and HCoV-OC43, mechanisms that may give rise to new variants capable of widespread intercontinental dissemination (PMID: 40295720). We highlight significant genetic discrepancies between circulating contemporary HCoV strains and classical laboratory-adapted reference strains, which may compromise the translational validity of in vitro studies, thereby emphasizing the necessity of updated reference systems that better represent currently circulating viruses (PMID: 40862615; PMID: 40638224). Additionally, we discuss concerns regarding the potential for increased pathogenicity in co-infections involving certain HCoVs, which may arise through synergistic modulation of the host immune response, competition for receptor binding, or altered cytokine dynamics, ultimately affecting disease progression and clinical outcomes (PMID: 39926579; PMID: 34932974; PMID: 38623185; PMID: 39891054; PMID: 31158256). Finally, we highlight concerns regarding the potential for genetic recombination and the emergence of novel viral variants and strains with increased pathogenicity and virulence (PMID: 35746710; PMID: 27774293). Integrating comparative genomics, molecular surveillance, and epidemiological data across seasonal HCoVs will therefore be essential to anticipate evolutionary trajectories, enhance the selection of laboratory reference strains, and guide the development of broad-spectrum medical countermeasures.
(2) Section 5. The conclusions, which span six pages, align more closely with the discussion section. It would be advisable for the authors to provide a concise summary of their work, no more than one page in length, in the section designated as "Conclusions." (Section 6).
Reply: We thank the reviewer for this kind suggestion. We fully agree that adding a dedicated "7. Conclusions" section would effectively summarize the key findings of the manuscript. Accordingly, we have restructured the content to ensure that the challenges and strategic approaches related to seasonal HCoVs are now discussed in the "6. Discussion" section. These revisions have been implemented in the revised manuscript. It should be noted that the Discussion is intended to be Section 6, which was incorrectly labeled as Section 5 in our previous manuscript. This error has now been corrected.
Reviewer 3 Report
Comments and Suggestions for Authors
The authors in this review, as they say in the abstract, want to examine the pathogenesis, epidemiology, genomic architecture, and major antigenic determinants of seasonal HCoVs, highlighting key differences in receptor usage and the roles of structural proteins in modulating viral tropism and host immunity. Moreover, they summarize recent advances across various vaccine platform, cross-reactive humoral and cellular immune responses following SARS-CoV-2 infection or vaccination, and the current challenges in pathogenesis research and vaccine development for seasonal HCoVs.
The Project is very ambitious and even if it is evident that the authors have read a lot of literature, the result is a weighted review that a researcher has difficulty reading it, struggling to understand what the authors want to highlight.
I therefore propose to the authors to streamline some parts to make it easier to read this review which has many interesting things to highlight.
First, the authors must rewrite the introduction by removing the part of the discovery of vaccines (Jenner) and Pasteur, as the 'history' is known to all biologists.
Instead, I would dwell on the role that SARS CoV vaccines have had on the pandemic and I would stress why the development of a vaccine against seasonal coronaviruses is so important, the evidence of a possible evolution towards a new pandemic and/or ....Perhaps it could be useful to move the lines 291-331 in the introduction, obviously rewriting everything more concisely.
When the authors describe the different seasonal viruses, they must follow the same logical pattern for all of them:
- When (year) when for the first time the virus was identified and type (alpha, beta...)
- Where did it originate from (bat...)
- Symptoms
- Structural protein and host cell receptor
In this context, it is not important how many days of incubation, the permissive cells and the mouse model are, this information can only be left in the figure.
To make it clear what a coronavirus looks like, it would be useful to make a figure representing the virus with the different structural proteins that constitute them and a cut part that shows the genome of the virus and the associated proteins.
Is it necessary, in this context to mention non-structural proteins (lines 352-374) in such depth?
Even when the authors describe the different types of vaccine, they must follow a pattern:
- Characteristic
- First time they have been used on humans and on what type of infectious disease
- Advantages
- Detriments
- There are vaccines authorized against different types of infectious disease (except for sarsCoV)
- There are vaccines authorized against SARSCoV, if they have since been withdrawn (ASTRAZENECA)
I would also move the order in which the vaccines are listed, starting from the 'classic' long-standing ones up to the mRNA ones. It would also be useful to mention the Cuban subunit vaccine. It would be useful to make a table with all the vaccines there are for SARS, the company, the efficacy, the side effects if they are still on the market.
In table 2, on the other hand, a column should be added specifying whether they are studies done on humans or mice.
The conclusions should also be lightened.
I hope I have been of help to you, I look forward to your changes
Minor revision
Generally VLP stands for Virus like particle and not for viral like particle
Author Response
Reviewer #3:
The authors in this review, as they say in the abstract, want to examine the pathogenesis, epidemiology, genomic architecture, and major antigenic determinants of seasonal HCoVs, highlighting key differences in receptor usage and the roles of structural proteins in modulating viral tropism and host immunity. Moreover, they summarize recent advances across various vaccine platform, cross-reactive humoral and cellular immune responses following SARS-CoV-2 infection or vaccination, and the current challenges in pathogenesis research and vaccine development for seasonal HCoVs.
The Project is very ambitious and even if it is evident that the authors have read a lot of literature, the result is a weighted review that a researcher has difficulty reading it, struggling to understand what the authors want to highlight.
Reply: We sincerely thank the reviewer for supporting our work and for offering the following insightful suggestions. We have carefully revised the entire manuscript in response.
I therefore propose to the authors to streamline some parts to make it easier to read this review which has many interesting things to highlight.
First, the authors must rewrite the introduction by removing the part of the discovery of vaccines (Jenner) and Pasteur, as the 'history' is known to all biologists.
Reply: We thank the reviewer for the kind suggestion. We agree with the reviewer that the discoveries of vaccines by Jenner and Pasteur are well established in the field of biology. Accordingly, we have removed the redundant historical details and streamlined the vaccine history section in the revised manuscript to enhance clarity and focus.
Instead, I would dwell on the role that SARS CoV vaccines have had on the pandemic and I would stress why the development of a vaccine against seasonal coronaviruses is so important, the evidence of a possible evolution towards a new pandemic and/or ....Perhaps it could be useful to move the lines 291-331 in the introduction, obviously rewriting everything more concisely.
Reply: We thank the reviewer for this thoughtful suggestion. We agree that emphasizing the importance of developing vaccines against seasonal HCoVs should be clearly stated in the introduction. As suggested, we have revised this section accordingly, removing the discussion on long COVID and case reports, and instead focusing more explicitly on the comparative analysis of symptoms among coronaviruses to better support the rationale for vaccine development. This revised section also has been moved to the penultimate paragraph of the Introduction.
When the authors describe the different seasonal viruses, they must follow the same logical pattern for all of them:
- When (year) when for the first time the virus was identified and type (alpha, beta...)
- Where did it originate from (bat...)
- Symptoms
- Structural protein and host cell receptor
Reply: We thank the reviewer for the thoughtful suggestions. We have carefully revised this section to ensure that all seasonal HCoVs are described in a consistent and logical structure. The reviewer recommended presenting the origins of seasonal HCoVs immediately after the discovery of each virus. However, we propose to slightly revise the current order, identification, clinical symptoms, origin, structural proteins, and host cell receptor usage, for the following reasons. Clinical symptoms were historically characterized alongside the initial identification of each virus; therefore, placing these two aspects together reflects their original scientific context and underscores their relevance to clinical diagnosis and vaccine development. In contrast, information on viral origins, structural proteins, and host cell receptors is largely derived from recent molecular epidemiological studies and advanced genomic analyses. Grouping these elements highlights the mechanisms of cross-species transmission and the molecular basis of human infection. Furthermore, evolutionary analyses of structural proteins depend on modern techniques and large-scale surveillance data, which are critical for guiding rational vaccine design. We respectfully request the reviewer's approval to retain this organizational framework, as it enhances both the scientific coherence and translational implications of the discussion.
In this context, it is not important how many days of incubation, the permissive cells and the mouse model are, this information can only be left in the figure.
Reply: We agree with the reviewer that the number of incubation days is not essential for our review; we have removed these details from the main text and retained them only in Table 1. However, permissive cell lines and mouse models are critical for the development of vaccines against seasonal HCoVs. In particular, suitable models remain limited for HCoV-HKU1. To highlight current challenges in seasonal HCoV vaccine development, we have summarized this information. Therefore, we kindly request the reviewer's permission to retain it.
To make it clear what a coronavirus looks like, it would be useful to make a figure representing the virus with the different structural proteins that constitute them and a cut part that shows the genome of the virus and the associated proteins.
Reply: We thank the reviewer for the kind suggestion. We have added Figure 1A to illustrate the virus with its distinct structural proteins (S, M, E, and HE), along with a cross-sectional view depicting the viral genome and associated N proteins. Figure 1B presents the genome organization of seasonal HCoVs, showing the overall genomic structures and corresponding open reading frames.
Is it necessary, in this context to mention non-structural proteins (lines 352-374) in such depth?
Reply: We thank the reviewer for highlighting this redundancy. We have streamlined the section in the revised manuscript to focus more specifically on antigenic structural proteins. Only seasonal HCoV-encoded enzymes are now briefly described, while detailed discussions of their roles in innate immune evasion have been removed to enhance clarity and conciseness.
Even when the authors describe the different types of vaccine, they must follow a pattern:
- Characteristic
- First time they have been used on humans and on what type of infectious disease
- Advantages
- Detriments
- There are vaccines authorized against different types of infectious disease (except for sarsCoV)
- There are vaccines authorized against SARSCoV, if they have since been withdrawn (ASTRAZENECA)
Reply: We thank the reviewer for the thoughtful suggestion. We agree that a consistent framework should be applied when describing different types of vaccines. In the revised manuscript, we first present the defining characteristics of each vaccine type, followed by an analysis of their respective advantages and limitations. The reviewer recommended including information on the initial use in humans and targeted infectious diseases immediately after characterizing each vaccine. While we appreciate this suggestion, we have chosen to address applications in a separate paragraph to maintain clarity and thematic coherence. Another reviewer noted that detailed accounts of historical development and non-coronavirus vaccines could be considered redundant in the context of our review, and suggested condensing these sections. In response, we have streamlined the historical background and instead focused the second paragraph of each vaccine type on conceptual evolution. Furthermore, we have retained and expanded upon successful examples of each platform against key respiratory viruses, including influenza, SARS-CoV-2, MERS-CoV, and RSV.
The reviewer noted that although numerous vaccines have been authorized against SARS-CoV-2, AstraZeneca withdrew its ChAdOx1-S-nCoV-19 vaccine globally last year. The company confirmed it had requested the European Medicines Agency (EMA) to revoke the vaccine's marketing authorization, a decision finalized in early May 2024. Similar actions have been taken in other regions. The withdrawal was driven by declining global demand, a market shift toward newer platforms such as mRNA vaccines, and strategic realignment of the company's product portfolio. Importantly, AstraZeneca emphasized that the decision was not related to the rare risk of thrombosis with thrombocytopenia syndrome (TTS) associated with the vaccine. This information, which is highly relevant, has been incorporated into subsection "4.5. Viral Vector Vaccine".
I would also move the order in which the vaccines are listed, starting from the 'classic' long-standing ones up to the mRNA ones. It would also be useful to mention the Cuban subunit vaccine. It would be useful to make a table with all the vaccines there are for SARS, the company, the efficacy, the side effects if they are still on the market.
Reply: We thank the reviewer for these thoughtful suggestions. We would like to respond to each comment in turn. First, the reviewer suggested reordering the vaccines listed, beginning with the classic, long-standing platforms and progressing to mRNA vaccines. While we agree that mRNA vaccines represent an innovative advancement in combating infectious diseases, we have positioned mRNA vaccines between DNA and subunit vaccines. Our rationale follows the inherent biological mechanisms of vaccine types based on the central dogma of molecular biology. Inactivated vaccines, representing whole-virus platforms, are classical and thus placed first. DNA vaccines function by entering cells, undergoing transcription into mRNA, and then translation into antigens. Similarly, mRNA vaccines bypass transcription and directly undergo translation into antigens upon delivery into target cells. In contrast, subunit and virus-like particle (VLP) vaccines present antigens directly, without requiring transcription or translation. Viral vector vaccines mimic natural infection by delivering genetic material into cells, which then express the antigen. Therefore, our ordering reflects a progression aligned with molecular biological principles, from nucleic acid-based platforms requiring gene expression to those delivering antigens directly. Presenting mRNA vaccines toward the end also highlights their technological advancement. Both organizational approaches, chronological development or functional mechanism, can effectively convey the evolution of vaccine platforms. We respectfully request the reviewer's approval to maintain our current structure. Second, the reviewer suggested including the Cuban Abdala subunit vaccine. We agree that this RBD-based subunit vaccine serves as a valuable example in illustrating the development and application of subunit vaccines. Accordingly, we have incorporated relevant references on the Cuban Abdala vaccine into Section "4.4. Subunit Vaccine" to strengthen the discussion and provide a more comprehensive overview. Third, the reviewer suggested including a table summarizing all SARS-CoV-2 vaccines, including the manufacturer, efficacy, side effects, and market availability. However, a comprehensive overview of SARS-CoV-2 vaccines falls outside the scope of our review, as our primary focus is on seasonal HCoVs. We only reference certain SARS-CoV-2 vaccines because some have demonstrated cross-reactivity with seasonal HCoVs, though this is not universally observed. Given that there are over 200 candidate SARS-CoV-2 vaccines, incorporating such a table would disproportionately shift the emphasis of our review. Therefore, we kindly request to maintain our focus on those SARS-CoV-2 vaccines that show evidence of cross-reactivity with seasonal HCoVs, aligning more closely with the central theme of this review.
In table 2, on the other hand, a column should be added specifying whether they are studies done on humans or mice.
Reply: We thank the reviewer for the insightful suggestion. We agree that it is important to clarify whether the cross-reactivity was assessed in human clinical trials, mice, non-human primates, or other animal models. We have carefully re-examined the cited studies and incorporated this critical information into Table 2. Additionally, we have used an asterisk (*) to denote neutralizing cross-reactivity, as vaccines exhibiting cross-neutralizing potential may effectively confer protection against target seasonal HCoVs.
The conclusions should also be lightened.
Reply: We agree with the reviewer that the Discussion section previously lacked clarity and conciseness. To address this, we have restructured the section to include two new subsections: "6.1. Current Challenges in Seasonal HCoV Vaccines" and "6.2. Strategic Framework for the Development of Seasonal HCoV Vaccines", which systematically summarize seven key challenges and three development strategies, respectively. Furthermore, a dedicated "7. Conclusions" section has been added to provide a concise and effective summary of the manuscript's key findings.
I hope I have been of help to you, I look forward to your changes.
Reply: We gratefully acknowledge the reviewer for their insightful comments and suggestions. In response, we have carefully revised the manuscript in its entirety to incorporate the recommended improvements.
Minor revision
Generally VLP stands for virus-like particle and not for viral-like particle.
Reply: We sincerely apologize for the error. After a thorough review, we have verified all spellings (including those in figures) and corrected "viral-like particle" to "virus-like particle," ensuring accuracy in terminology throughout the manuscript.
Round 2
Reviewer 1 Report
Comments and Suggestions for Authors
Dear Authors, you did a great revision!
Particularly Strong Improvements:
- Antigenic sin discussion (Lines 1264-1290): Your mechanistic explanation of conflicting cross-reactivity data is outstanding and will become a frequently cited framework for the field.
- Section 2.6 (Public Health Rationale): This new section perfectly addresses the "why vaccinate against mild infections" question with the varicella analogy and herd immunity considerations.
- Table 2: The asterisk notation distinguishing binding versus neutralizing cross-reactivity is elegant and highly useful.
- Animal model limitations: Your discussion of HCoV-HKU1 challenges (Lines 308-321, 1177-1202) demonstrates appropriate scientific rigor while clearly identifying critical research gaps.
- Quantitative data: The manuscript now provides specific titers, fold-changes, and statistical measures throughout, greatly enhancing its utility as a reference.
The literature search strategy, corrected inconsistencies, and comprehensive citations all strengthen the manuscript's scholarly foundation. This review will serve as the authoritative reference on seasonal HCoV vaccines that the field needs.
Congratulations on excellent work.
Reviewer 3 Report
Comments and Suggestions for Authors
None